# On the identifiability of causal graphs with the invariance principle

**Francesco Montagna**
Institute of Science and Technology Austria, Chan Zuckerberg Initiative
`francesco.montagna@ist.ac.at`

## Abstract

Causal discovery from i.i.d. observational data is known to be generally ill-posed. We demonstrate that if we have access to the distribution induced by a structural causal model, and additional data from *only two* environments with invariant causal mechanisms and sufficiently different noise statistics, the unique causal graph is identifiable. Notably, this is the first result in the literature that guarantees the entire causal graph recovery with a constant number of environments and arbitrary nonlinear mechanisms. Our only constraint is the Gaussianity of the noise terms; however, we propose potential ways to relax this requirement. Of interest on its own, we expand on the well-known duality between independent component analysis (ICA) and causal discovery; recent advancements have shown that nonlinear ICA can be solved from multiple environments, at least as many as the number of sources: we show that the same can be achieved for causal discovery while having access to much less auxiliary information.

## 1 Introduction

Causal discovery seeks to recover cause–effect structure from data, which allows counterfactual reasoning and prediction under interventions (Pearl, 2009; Peters et al., 2017; Spirtes, 2010; Spirtes et al., 2000). However, learning causal structure from *purely observational* i.i.d. data is, in general, ill-posed: multiple directed acyclic graphs (DAGs) are distributionally equivalent, i.e., indistinguishable from the data distribution.

Recent work has explored the problem of causal graph identifiability from multiple environments and soft interventions (i.e., in the setting where non i.i.d. data might naturally occur and does not stem from changes in the causal structure) (Perry et al., 2022; Huang et al., 2020; Heinze-Deml et al., 2018; Peters et al., 2015; Ghassami et al., 2017; 2018; Jaber et al., 2020; Jalaldoust et al., 2025; Brouillard et al., 2020; Heurtebise et al., 2025): however, from an identifiability perspective, these results do not provide guarantees of recovery of the unique causal graph with a limited number of environments under generic assumptions.

Our research overcomes this limitation. We prove that, for structural causal models (SCMs) with arbitrary nonlinear mechanisms, auxiliary information from *only two* sufficiently distinct environments is enough to identify the unique causal graph. Our only constraint is the Gaussianity of the noise terms; however, we outline potential ways to relax this requirement. To our knowledge, this is the first proof of identifiability for full graphs of arbitrary size and generic functional mechanisms from a constant number of environments. Strengthening our findings is the contrast with hard-intervention regimes, where state-of-the-art theory requires the number of experiments to scale with the number of nodes (Eberhardt et al., 2005)).

Our work is also of independent methodological interest. In particular, our contributions are built on the duality between causal discovery and independent component analysis (ICA). First Monti et al. (2020), and later Reizinger et al. (2023) recently formalized that nonlinear ICA identifiability results naturally extend to structure learning (well known in the linear case since Shimizu et al. (2006)). This is of great relevance in light of the late advancements in multi-environment ICA identifiability pioneered by Hyvärinen & Morioka (2016); however, directly bootstrapping these findings to causal discovery doesn't carry great promise, being ICA the harder problem of the two: we show that where ICA identifiability requires a number of environments that scales linearly with the number

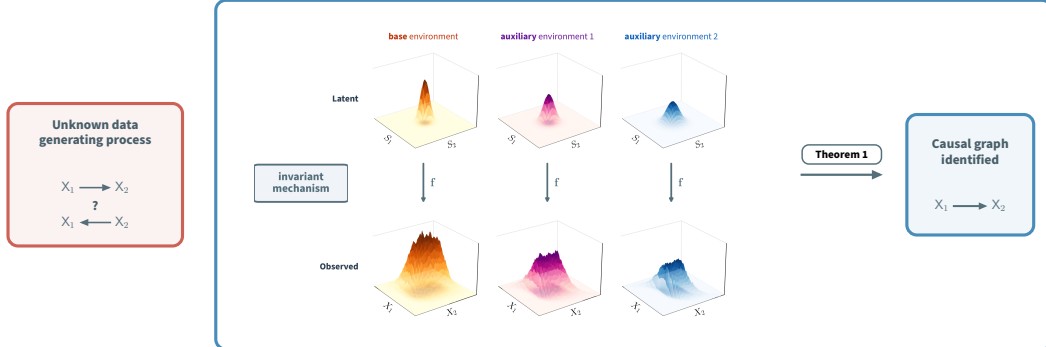

Figure 1: Given $X_1 := S_1$, $X_2 := f(X_1, S_2)$, the causal direction cannot be inferred from i.i.d. data alone (red box, left). Multi-environment data with invariant mechanisms $f$ but different latent noise distributions adds the missing constraints: Theorem 1 shows that additional data from just 2 auxiliary environments suffice to uniquely identify the causal graph for arbitrary size and nonlinear mechanisms. Multi-environment data with invariant mechanisms has already proved useful for recovering causal structure in gene regulatory networks (Meinshausen et al., 2016), providing a concrete motivating example for our theory.

of variables, causal graph identifiability can be achieved with data from just two extra domains. This calls for causality-only identifiability results in the multi-environment setting, as developed in our work. Inspired by the recent success of ICA with multiple environments, we are hopeful that our approach paves the way to novel causality theory that weakens the requirements in terms of heterogeneity of the data and parametric assumptions.

Our main contribution is as follows:

> 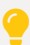 We show that the acyclic causal graph underlying an *arbitrary invertible structural causal model* with Gaussian noise terms is identifiable from *only two* sufficiently different auxiliary environments (Figure 1).

Moreover:

- We introduce proof techniques that are novel for causal discovery and leverage the (well-known) duality between structural causal models and independent component analysis; to the best of our knowledge, these are the first causality-only identifiability results for nonlinear SCMs that stem from this connection.
- We empirically validate our theory through synthetic experiments [1].

**Main related works.** Causal discovery with multiple environments and linear SCMs was popularized by Peters et al. (2015); Heinze-Deml et al. (2018) extended their results to nonlinear additive noise models. Rothenhäusler et al. (2015) is the closest to our paper, but their results are limited to linear models. The most relevant reference from the ICA literature is Hyvärinen & Morioka (2016), which are the first to illustrate how multiple environments with invariant mechanisms unlock identifiability for ICA models with arbitrary invertible nonlinear mixing functions. A thorough treatment of the literature relevant to our paper is found in Section C.

## 2 PRELIMINARIES

First, we define structural causal models, independent component analysis, and how they relate. Then, we describe the problem of causal discovery from multiple environments and define identifiability of causal graphs in this context.

---

[1] https://github.com/francescomontagna/gaussian-multienv-cd.git

## 2.1 STRUCTURAL CAUSAL MODELS AND ICA

Let us consider a set of causal variables $\mathbf{X}$, with components generated according to a structural causal model

$$X_i := F_i(\mathbf{X}_{\mathrm{PA}_i}, S_i), \quad \forall i = 1, ..., d, \tag{1}$$

where $\mathbf{X}_{\mathrm{PA}_i}$ are the causes of $X_i$, specified by a directed acyclic graph (DAG) $\mathcal{G}$ with nodes $\mathbf{X}$. $\mathrm{PA}_i \subset \{1, ..., d\}$ denotes the indices of the parents of $X_i$ in the graph. The functions $F_i$ are the *causal mechanisms* that map causes to effects. We assume mutually independent noise terms $\mathbf{S} = (S_1, ..., S_d)$ with density $p_\theta$, where $\theta$ is a set of parameters defining the density function. Further, we restrict to structural causal models where there are no latent common causes.

It is well known that the SCM of Equation (1) can be expressed in the form of an ICA model $(\mathbf{f}, p_\theta)$:

$$\mathbf{X} = \mathbf{f}(\mathbf{S}), \qquad p_\theta(\mathbf{s}) = \prod_{i=1}^{d} p_{i,\theta}(s_i), \tag{2}$$

where $\mathbf{f}$ is the ICA *mixing function*, uniquely specified by the SCM (see Section H.2).

**Notational remarks.** Uppercase letters (e.g., $\mathbf{S}$) denote random variables, lower case letters (e.g., $\mathbf{s}$) their realizations. Bold letters are reserved for vectors and vector-valued functions. For an integer $k$, $[k] := \{1, ..., k\}$. Further, we define the *support* to keep track of the nonzero entries in matrices: for a matrix $M$, $\mathrm{supp}(M) := \{(i,j)|i \in [m], j \in [n]$ and $M_{ij} \neq 0\}$; for a matrix valued function $M$ the support is defined as $\mathrm{supp}(M) = \{(i,j)|i \in [m], j \in [n]$ and there is $\mathbf{x}$ s.t. $M_{ij}(\mathbf{x}) \neq 0\}$.

It is known (Reizinger et al., 2023) that, under some *faithfulness* assumption, the support of the Jacobian of the mixing function completely identifies the causal structure.

**Definition 1** (Faithfulness). *Consider* $\mathbf{x} = \mathbf{f}(\mathbf{s})$. *We say that* $J_{\mathbf{f}^{-1}}(\mathbf{x})$ *is* faithful *if for each* $i, j \in [d]$ $J_{\mathbf{f}^{-1}}(\mathbf{x})_{ij} = 0 \iff S_i$ *is constant in* $X_j$ *on the entire domain. In other words:*

$$\mathrm{supp}(J_{\mathbf{f}^{-1}}(\mathbf{x})) = \mathrm{supp}(J_{\mathbf{f}^{-1}}). \tag{3}$$

**Proposition 1** (Proposition 1 in Reizinger et al. (2023)). *Let* $J_{\mathbf{f}^{-1}}(\mathbf{x})$ *faithful. Then, for each* $i \neq j$:

$$J_{\mathbf{f}^{-1}}(\mathbf{x})_{ij} = 0 \iff j \notin \mathrm{PA}_i.$$

This formulation of faithfulness is well known and at the core of the LiNGAM algorithm for linear SCMs (Shimizu et al., 2006), and is satisfied almost everywhere under some regularity conditions on $\mathbf{f}$. When this is the case, the above proposition means that for causal discovery we are interested in the support of the inverse Jacobian, and, by Equation (3), this can be recovered by having access to the support at a single point where faithfulness is satisfied.

Next, we introduce the notion of *environment* and define the causal discovery problem when multiple environments are available.

## 2.2 THE INVARIANCE PRINCIPLE AND NOTIONS OF IDENTIFIABILITY

Consider the ICA model of Equation (2). We define an *environment* as the pair $(\mathbf{f}, p_\theta^e)$, where:

$$\mathbf{X} = \mathbf{f}(\mathbf{S}), \qquad p_\theta^e(\mathbf{s}) = \prod_{i=1}^{d} p_{i,\theta}^e(s_i). \tag{4}$$

Superscript indices are reserved to specify the environment. The key feature is that, compared with Equation (2), the mixing function $\mathbf{f}$ is invariant, while we allow changes in the density of the sources.

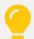 **The invariance principle.** Given a base environment $(\mathbf{f}, p_\theta)$, an auxiliary environment is characterized by invariant causal mechanism $\mathbf{f}$ and shifts in the source density $p_\theta^e \neq p_\theta$.

From a notational point of view, the *base* environment is denoted with index 0, i.e. $p_\theta^0 := p_\theta$; *auxiliary* environments will appear with index $e \in \mathcal{E}$, where $\mathcal{E} \subset \mathbb{N}_{>0}$ is the set of auxiliary environment

indices. Real-world examples in the causality literature where the invariance principle is satisfied can be found in Section H.5.

Intuitively, causal discovery is the inference problem of finding the causal graph underlying a structural causal model from the data. We are interested in causal discovery from multiple environments. Identifiability is achieved when the graph underlying the structural causal model is uniquely specified by the causal variables' distribution. In the definition, we denote the pushforward of a density $p$ by $\mathbf{f}$ with $\mathbf{f}_* p$.

**Definition 2** (Identifiability of the causal graph). *Consider a structural causal model* $(\mathbf{f}, p_\theta^0)$*, and* $(\mathbf{f}, p_\theta^e)_{e \in \mathcal{E}}$ *auxiliary environments. We say that the causal graph underlying the SCM is identifiable if, given an alternative model* $(\widehat{\mathbf{f}}, p_{\widehat{\theta}}^0)$ *with auxiliary environments* $(\widehat{f}, p_{\widehat{\theta}}^e)_{e \in \mathcal{E}}$*, then:*

$$\mathbf{f}_* p_\theta^e = \widehat{\mathbf{f}}_* p_{\widehat{\theta}}^e \quad \forall e \in \{0\} \cup \mathcal{E} \implies \mathrm{supp}(J_{\mathbf{f}^{-1}}) = \mathrm{supp}(J_{\widehat{\mathbf{f}}^{-1}}).$$

The above definition of identifiability, based on the support of the Jacobian inverse of the mixing function, may be a bit unfamiliar, but it's equivalent to what is commonly meant when asking that a causal DAG is identifiable:

> 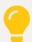 Any alternative causal model that matches the distribution of the data is compatible only with the ground truth causal graph (represented with the inverse Jacobian's support).

**Relation with ICA identifiability.** Compare Definition 2 of identifiability of the causal graph with the notion of identifiability in ICA of Definition 3 in the appendix: for causal discovery, all we care about is the support of $J_{\mathbf{f}^{-1}}$, which can be identified from any point where the Jacobian is faithful; for independent component analysis, we need to guarantee that the exact values of the Jacobian can be recovered over each point of the domain, up to trivial indeterminacies. This phrasing clarifies that, in the nonlinear setting (where the Jacobian varies with $\mathbf{x}$), causal discovery is a much simpler problem than ICA: it only requires identifying the support at a single point, rather than the value at any point. This is reflected in our main identifiability result (Theorem 1): we will show that the causal graph of a nonlinear SCM can be identified with the information from only two auxiliary environments; this in stark contrast with ICA identifiability results for general mixing functions, that usually require a number of environments that scales linearly ($\mathcal{O}(d)$) with the number of sources.

> **Problem definition.** We aim to characterize the conditions under which the causal graph $\mathcal{G}$ is identifiable from the fewest possible environments.

## 3 THEORY

To develop our theory, we rely on the following assumptions on the ICA model of Equation (2).

**Assumption 1** (Invertibility). $\mathbf{f}$ *is a global diffeomorphism and twice differentiable.*

**Assumption 2** (Rescaling environments). *Each environment is obtained as a rescaling of* $\mathbf{S}$*, namely* $\mathbf{S}^e$ *is distributionally equivalent to* $L_e \mathbf{S}$ *for each* $e \in \mathcal{E}$*, with* $L_e = \mathrm{diag}(\lambda_1^e, \dots, \lambda_d^e)$*. We ask that for each* $j \in [d]$ *there is at least one* $e \in \mathcal{E}$ *such that* $\lambda_j^e \neq 0$*.*

**Assumption 3** (Faithfulness). *For* $\mathbf{f}^{-1}(\mathbf{x}) = \mathbf{s}$ *where* $\mathbf{s} = \mu_{\mathbf{S}}$*, the mean of the vector of sources, the Jacobian is faithful (Definition 1).*

**Assumption 4** (Gaussianity). $\mathbf{S}$ *has Gaussian density* $p_\theta$ *with* $\theta$ *mean and covariance matrix parameters.*

**Discussion on the Assumptions 1-4.** Assumption 1 is standard when proving identifiability: the results in Hoyer et al. (2008); Zhang & Hyvärinen (2009); Immer et al. (2022) are based on higher-order derivatives, and have strong requirements that guarantee diffeomorphic causal mechanisms (Corollary 3.5 in (Dominguez-Olmedo et al., 2023)). Also Assumption 2 is mild and somewhat necessary: it simply asks that the interventions are *meaningful*, i.e. that they affect the variance;

interventions on the mean, intuitively, are not informative as they shift the density graph by a constant, without affecting its *shape* (the gradient and the Hessian of the density, where information about the causal graph lies). Assumption 3 requires that the Jacobian of the inverse of the mixing function is informative about the causal structure at the mean of $\mathbf{S}$ (and it's almost surely verified over $\mathbf{X}$ samples, under some generic regularity conditions on $\mathbf{f}$). The reason behind it is that we probe the identifiability of the Jacobian's support at the mean. The only real simplifying constraint is Assumption 4 of the Gaussianity of the sources, which is, however, not new in the literature (see, e.g., Rolland et al. (2022)). Later, we discuss why this assumption is needed in the paper and potential ways to relax it (Section 3.1).

In the remainder of the paper we demonstrate that, under these assumptions, leveraging the ICA formalism we can prove the identifiability of causal graphs, potentially with as few as two auxiliary environments. Our starting point is the invertibility $\mathbf{f}$, so that we can write the density of $\mathbf{X}$ with the change of variable for each value $\mathbf{x} = \mathbf{f}(\mathbf{s})$ as:

$$p(\mathbf{x}) = p_\theta(\mathbf{s})|J_{\mathbf{f}^{-1}}(\mathbf{x})|. \tag{5}$$

Consider an alternative invertible ICA model (Equation (2)) $(\widehat{\mathbf{f}}, p_{\hat\theta})$ such that:

$$p(\mathbf{x}) = p_{\hat\theta}(\mathbf{s})|J_{\widehat{\mathbf{f}}^{-1}}(\mathbf{x})|. \tag{6}$$

We define the *indeterminacy function*

$$\mathbf{h} := \widehat{\mathbf{f}}^{-1} \circ \mathbf{f}, \tag{7}$$

which "quantifies" how different the two ICA solutions are. By the multivariate chain rule, the following relation among Jacobian matrices holds:

$$J_{\mathbf{f}} = J_{\widehat{\mathbf{f}}} J_{\mathbf{h}}. \tag{8}$$

We show that (under Assumptions 1-4 on $(\mathbf{f}, p_\theta)$) there is at least one point $\mathbf{x} = \mathbf{f}(\mathbf{s}) = \widehat{\mathbf{f}}(\hat{\mathbf{s}})$ such that the Jacobian $J_{\mathbf{h}}(\mathbf{s})$ is a scaled permutation, meaning that $J_{\mathbf{f}^{-1}}$ support is identifiable up to column permutation. Given that for acyclic causal models permutations are easily removed (Shimizu et al., 2006), this is equivalent to identifiability of the causal graph in the sense of Definition 2, as we discuss next.

### 3.1 IDENTIFIABILITY FROM SECOND ORDER DERIVATIVES OF THE LOG-LIKELIHOOD

In this section, we present our main theoretical result and the intuitions behind it. Our argument for identifiability relies on the analysis of the Hessian of the log-likelihood of $\mathbf{X}^e$ for all environments. We consider the case where $\mathbf{f}^{-1}(\mathbf{x}) = \mathbf{s} = \mu_{\mathbf{S}}$ (by construction, there is a unique corresponding $\hat{\mathbf{s}} = \hat{\mathbf{f}}^{-1}(\mathbf{x})$). We partition the set of auxiliary environments $\mathcal{E}$ into two groups $\mathcal{E}_1$ and $\mathcal{E}_2$. Then, we define the following quantities:

$$
\begin{aligned}
\Omega_1 &:= \sum_{e \in \mathcal{E}_1} D_{\mathbf{s}}^2 \log p_\theta(\mathbf{s}) - D_{\mathbf{s}}^2 \log p_\theta^e(\mathbf{s}) \\
\Omega_2 &:= \sum_{e \in \mathcal{E}_2} D_{\mathbf{s}}^2 \log p_\theta(\mathbf{s}) - D_{\mathbf{s}}^2 \log p_\theta^e(\mathbf{s}),
\end{aligned} \tag{9}
$$

where $D^2$ denotes the differential operator that returns the Hessian matrix. Similarly, we define $\widehat{\Omega}_1, \widehat{\Omega}_2$ by replacing $\theta$ with $\hat\theta$. The introduction of $\Omega_l, \widehat{\Omega}_l, l = 1, 2$, is instrumental for the next result.

**Lemma 1.** *Let $\mathbf{x} = \mathbf{f}(\mathbf{s}) = \widehat{\mathbf{f}}(\hat{\mathbf{s}})$, where $\mathbf{s} = \mu_{\mathbf{S}}$. Let Assumptions 1,2 and 4 satisfied. Then:*

$$\sum_{e \in \mathcal{E}_1} D_{\mathbf{x}}^2 \log p(\mathbf{x}) - D_{\mathbf{x}}^2 \log p^e(\mathbf{x}) = J_{\mathbf{f}^{-1}}(\mathbf{x})^T \Omega_1 J_{\mathbf{f}^{-1}}(\mathbf{x}) = J_{\widehat{\mathbf{f}}^{-1}}(\mathbf{x})^T \widehat{\Omega}_1 J_{\widehat{\mathbf{f}}^{-1}}(\mathbf{x}) \tag{10}$$

$$\sum_{e \in \mathcal{E}_2} D_{\mathbf{x}}^2 \log p(\mathbf{x}) - D_{\mathbf{x}}^2 \log p^e(\mathbf{x}) = J_{\mathbf{f}^{-1}}(\mathbf{x})^T \Omega_2 J_{\mathbf{f}^{-1}}(\mathbf{x}) = J_{\widehat{\mathbf{f}}^{-1}}(\mathbf{x})^T \widehat{\Omega}_2 J_{\widehat{\mathbf{f}}^{-1}}(\mathbf{x}) \tag{11}$$

The proof is derived by direct computation and can be found in Section D.2. We point to Lemma 7 in Varici et al. (2025) for related results that analyze the difference of first-order derivatives of the log-likelihood, in the context of causal representation learning with soft interventions.

We can intuitively illustrate how the identifiability of the Jacobian's support follows from our Lemma 1. A first remark is that the $\Omega_l, \widehat{\Omega}_l$ matrices are diagonal. That is because, for a vector of mutually independent random variables, the Hessian of the log-density is diagonal (see Section H.3 for details about it). Second, by the chain rule, Equations (10) and (11) imply $J_{\mathbf{h}}(\mathbf{s})^T \widehat{\Omega}_l J_{\mathbf{h}}(\mathbf{s}) = \Omega_l$ for $l = 1, 2$, from which

$$J_{\mathbf{h}}(\mathbf{s})^{-1} \widehat{\Omega}_1^{-1} \widehat{\Omega}_2 J_{\mathbf{h}}(\mathbf{s}) = \Omega_1^{-1} \Omega_2. \tag{12}$$

This means that $J_{\mathbf{h}}(\mathbf{s})$ maps one diagonal matrix to another: if the eigenvalues of $\widehat{\Omega}_1^{-1} \widehat{\Omega}_2$ are distinct, that is enough to force $J_{\mathbf{h}}(\mathbf{s})$ to a scaled permutation, which is exactly our goal. This sketched argument is key to understanding how Equations (10) and (11) provide enough constraints to identify the support of $J_{\mathbf{f}^{-1}}$. Clearly, this discussion implicitly requires that $\Omega_l$ and $\widehat{\Omega}_l$ are full rank. This can be achieved under the following conditions over the rescaling matrices $L_e = \text{diag}(\lambda_1^e, \ldots, \lambda_d^e)$ that define the multiple environments.

**Assumption 5** (Sufficient variability). *For each $j \in [d]$:*

$$\sum_{e \in \mathcal{E}_1} \frac{1}{(\lambda_j^e)^2} \neq |\mathcal{E}_1| \quad \text{and} \quad \sum_{e \in \mathcal{E}_2} \frac{1}{(\lambda_j^e)^2} \neq |\mathcal{E}_2|.$$

The assumption basically requires that there is sufficient variability between the different environments. Similar requirements of sufficient variability are ubiquitous in the nonlinear ICA literature (e.g. Hyvärinen & Morioka (2016); Khemakhem et al. (2020b); Lachapelle et al. (2022)). Intuitively speaking, Assumption 5 is satisfied when, for each of the two groups of environments ($\mathcal{E}_1$ and $\mathcal{E}_2$), each source $S_j$ is subject to rescaling. To see that, consider the LHS of the first equation: $\lambda_j^e = 1$ for each $e \in \mathcal{E}_1$ corresponds to the case when the variable $S_j$ is never subject to rescaling in any of the environments, and indeed yields a violation of the assumption. Note that even if $S_j$ is subject to rescaling for some $e \in \mathcal{E}_1$, the values of $(\lambda_j^e)_{e \in \mathcal{E}_1}$ can always be tuned such that the assumption is violated; however, this corresponds to pathological choices of the rescaling coefficients, which never occur in general (shown in Proposition 3 in the appendix).

Next, we are ready to state our main identifiability result.

**Theorem 1.** *Consider the groundtruth ICA model $(\mathbf{f}, p_\theta)$ of Equation (2) and the alternative $(\widehat{\mathbf{f}}, p_{\hat{\theta}})$. Let Assumptions 1-5 be satisfied, and assume that the elements in the set $\{(\Omega_1^{-1}\Omega_2)_{ii}\}_{i=1}^d$ are pairwise distinct. Let $\mathbf{x} = \mathbf{f}(\mathbf{s}) = \widehat{\mathbf{f}}(\hat{\mathbf{s}})$ and $\mathbf{s} = \mu_{\mathbf{S}}$: then, the indeterminacy function $\mathbf{h} := \widehat{\mathbf{f}}^{-1} \circ \mathbf{f}$ satisfies $J_{\mathbf{h}}(\mathbf{s})$ full rank and diagonal, meaning that the causal graph $\mathcal{G}$ is identifiable.*

Theorem 1 assumes that the elements in the set $\{(\Omega_1^{-1}\Omega_2)_{ii}\}_{i=1}^d$ are pairwise distinct. This requirement excludes pathological choices of the coefficients of the rescaling matrices $L_e$ that define the multiple environments, and it is generically satisfied (Proposition 4 in the appendix).

*Proof sketch (full proof in Section D.4).* By Lemma 1 we have

$$M^T \Omega_l M = \widehat{\Omega}_l, \quad l = 1, 2, \tag{13}$$

where $M := J_{\mathbf{h}^{-1}}(\hat{\mathbf{s}})$. Define $A := \widehat{\Omega}_1^{-1}\widehat{\Omega}_2$ and $B := \Omega_1^{-1}\Omega_2$. From Equation (13) we can show that $A = M^{-1}BM$, i.e. that $A$ and $B$ are similar. Moreover, being $\{(\Omega_1^{-1}\Omega_2)_{ii}\}_{i=1}^d$ elements pairwise distinct, the diagonal elements of $A$ and $B$ are never repeated. Note that the eigenvectors of a diagonal matrix with all distinct eigenvalues are aligned with the standard basis: given that $M$, by definition of similarity, maps the eigenvectors of $A$ to eigenvectors of $B$, we conclude that it is a scaled permutation. The permutation is removed leveraging the acyclicity of the causal model, according to Lemma 1 in Reizinger et al. (2023). Assumption 3 implies that the causal graph is identified. $\square$

**Identifiability from two auxiliary environments.** The theorem tells that, given that we have access to two groups of auxiliary environments, both inducing changes in the variance of all sources, at the mean of the sources the ground truth and the alternative models are equivalent up to rescaling. This constrains the support of $J_{\widehat{\mathbf{f}}^{-1}}$ of the alternative model to be equal to that of $J_{\mathbf{f}^{-1}}$, which is enough to guarantee identifiability of the causal graph. It is interesting to discuss the theorem when $|\mathcal{E}| = 2$, showing that the above result demonstrates identifiability with as few as two additional environments. In this setting, with $\mathcal{E}_1 = \{1\}$, $\mathcal{E}_2 = \{2\}$, if $L_1 = \text{diag}(\lambda_j^1)_{j=1}^d$ and $L_2 = \text{diag}(\lambda_j^2)_{j=1}^d$ with

$\lambda_j^1, \lambda_j^2 \neq 1$ for each $j \in [d]$, then we have two extra environments where the variance of *all* the sources is affected by rescaling. This is sufficient to guarantee that the assumptions of Theorem 1 are met. An important consequence is that the number of required environments does not scale with the number of nodes in the graph, in contrast with similar findings for nonlinear ICA identifiability. As long as there is sufficient variability in the sources of two environments (relative to the base model), we are always guaranteed that the causal graph can be recovered.

**Theorem 1 beyond Gaussianity.** Theorem 1 inherits the assumption of Gaussianity from Lemma 1; here, we briefly discuss potential ways to relax it. At a general point $\mathbf{x} = \mathbf{f}(\mathbf{s})$ the Hessian of the log-likelihood is equal to

$$J_{\mathbf{f}^{-1}}(\mathbf{x})^T D_{\mathbf{s}}^2 \log p^e(\mathbf{s}) J_{\mathbf{f}^{-1}}(\mathbf{x}) + D_{\mathbf{x}}^2 \log |J_{\mathbf{f}^{-1}}(\mathbf{x})| + \sum_{j=1}^{d} \partial s_j \log p^e(s_j) D^2 \mathbf{f}_j^{-1}(\mathbf{x}).$$

The log-determinant term cancels by taking the difference between environments. To recover Equations (10) and (11) in Lemma 1, we note that the summation of second-order derivatives vanishes when $\nabla \log p^e(\mathbf{s}) = 0$, namely at the mean of the Gaussian sources. However, this can hold for any source distribution that has at least one point where the gradient is zero, a remark that naturally extends Lemma 1 (and hence, Theorem 1) to a larger class of causal models. Moreover, from a practical perspective, even if the gradient of the log-likelihood of the sources does not vanish, Lemma 1 is *approximately* true when the gradient is sufficiently small. This can occur, e.g., for heavy-tailed distributions. This analysis should convince that Gaussianity is a sufficient but not necessary requirement, and hopefully inspire future research to extend our identifiability results. Mathematical details on the steps in this paragraph, as well as an expanded discussion on the generalization of our theory for more general classes of distributions, are found in Section G.1.

Next, we validate the conclusions of Theorem 1 with experiments.

## 4 EMPIRICAL RESULTS

In this section, we report and analyse empirical results that validate our theory. Our experiments on synthetic data show that if the assumptions of Theorem 1 hold, the causal direction can be recovered from the data. In the main paper, we focus on bivariate graphs, commonly adopted as the easiest yet non-trivial setting for testing identifiability (e.g., Hoyer et al. (2008); Zhang & Hyvärinen (2009); Immer et al. (2022)). Additional experiments on multivariate causal graphs are in Section F.5.

### 4.1 SYNTHETIC DATA GENERATION

We generate synthetic data from bivariate causal models with independent noise terms, sampled from a normal distribution with unit mean and covariance entries uniformly drawn between $[1, 1.5]$. Given the variables $x_1, x_2$ and the graph $x_1 \to x_2$ we consider the following causal mechanisms that comply with the assumptions of Theorem 1: *(i)* $x_2 := s_1^2 \arctan(s_2) + s_2^3$ *(ii)* $x_2 := s_1^2 s_2 + \arctan(s_2)$ *(iii)* $x_2 := s_1^2 + \arctan(s_1) s_2 + s_1 s_2^3$. Note that any of these models can not be reparametrized to a post nonlinear or location scale noise model, which are the most general SCMs identifiable from pure observations (Zhang & Hyvärinen, 2009; Immer et al., 2022). Additionally, we consider data from a linear Gaussian model, notably non-identifiable. We run experiments on datasets with $\{3, 6, 9\}$ environments. For each environment, we generate 2000 observations. In Section F.4, we discuss experiments with non-Gaussian independent sources. Interestingly, these additional results seem to support our hypothesis that Theorem 1 could be extended to other source distributions.

### 4.2 ANALYSIS OF THE EXPERIMENTAL RESULTS

In this section, we analyse the empirical results. First, we introduce an algorithm for inferring the Jacobian support that leverages our theory.

**Algorithm.** The simplified pseudocode is found in Algorithm 1 (a detailed version is presented in Section F.3). The steps in our procedure closely follow the proof of Theorem 1: this approach to algorithmic design is not necessarily the best, which is why we highlight that our method is not

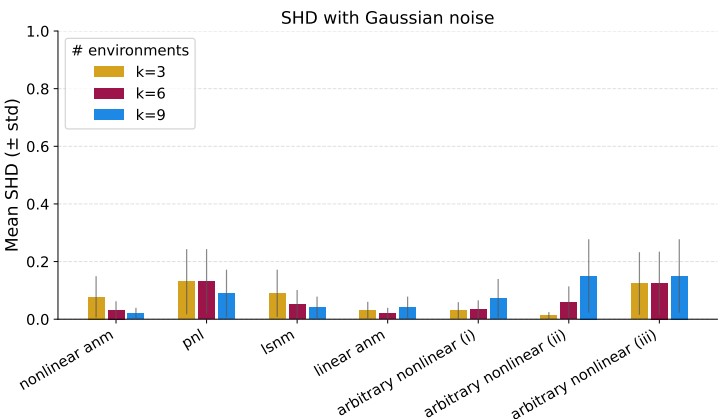

Figure 2: Average SHD (0 is best, 1 is worst) achieved by Algorithm 1 over 50 seeds on binary graphs. When the assumptions of Theorem 1 are satisfied, the method can appropriately infer the causal direction, both in the observationally identifiable setting (nonlinear ANM, PNL, LSNM) and the observationally non-identifiable one (linear Gaussian model and the three SCMs with arbitrary nonlinearity). The number of environments does not have a notable effect on the accuracy.

---

**Algorithm 1:** Estimating supp $J_{\mathbf{f}^{-1}}$ from the data (algorithm sketch)

---

**Data:** $\widehat{X} \in \mathbb{R}^{k \times n \times d}$                `// ∀ env:  n d-dimensional observations.`
      $\mathcal{E}_1, \mathcal{E}_2 \subset [k]$       `// Set of indices splitting the environments in two groups`

**Result:** Estimate of supp $J_{\mathbf{f}^{-1}}$

$\widehat{S} \leftarrow \text{score\_estimate}(\widehat{X}) \in \mathbb{R}^{k \times n \times d}$

$\widehat{H} \leftarrow \text{hess\_estimate}(\widehat{X}) \in \mathbb{R}^{k \times n \times d \times d}$

`// For each environment e, find the sample corresponding to the mean of the source`

**for** $e = 1, ..., k$ **do**
    $m_e \leftarrow i$ s.t. $\mathbf{f}^{-1}(\widehat{X}[e, i]) \approx \mu_{\mathbf{S}}$
**end**

`// Difference of Hessians at the mean (i.e.  Equations (10) and (11))`

$\widehat{H}_{\text{diffs}} \leftarrow 0 \in \mathbb{R}^{2 \times d \times d}$

**for** $\ell = 1, 2$ **do**
    **for** $e \in \mathcal{E}_\ell$ **do**
        $\Delta_H = \widehat{H}[0, m_1] - \widehat{H}[e, m_e]$       `// m_1 is the index for the base environment`
        $\widehat{H}_{\text{diffs}}[\ell] \leftarrow \widehat{H}_{\text{diffs}}[\ell] + \Delta_H.$
    **end**
**end**

$M \leftarrow \widehat{H}_{\text{diffs}}^{-1}[1] \widehat{H}_{\text{diffs}}[2] \approx J_{\mathbf{f}} \Omega_1^{-1} \Omega_2 J_{\mathbf{f}^{-1}}$    `// H_diffs[ℓ] ≈ J_{f^-1}^T Ω_ℓ J_{f^-1}, by Equations (10) and (11)`

$\widehat{J}_{\mathbf{f}^{-1}} \leftarrow \text{diagonalize}(M) \approx J_{\mathbf{f}^{-1}} D P$

**return** supp $\left( \widehat{J}_{\mathbf{f}^{-1}} P^{-1} \right)$       `// P can be found using the acyclicity of the causal graph.`

---

within our main contributions. For a single inference, the input is the data tensor $\widehat{X} \in \mathbb{R}^{k \times n \times d}$: for each environment from 1 to $k$ it consists of a dataset with $n$ observations of $d$ causal variables. Additionally, we are given the sets $\mathcal{E}_1, \mathcal{E}_2 \subset [k]$ of indices that split the auxiliary environments into two groups, as required by our theory. The first environment is taken as the base one. We have two steps where statistical estimation is involved: *(i)* For each environment, the gradient and the Hessian of the log-likelihood are approximated via the Stein gradient estimator, introduced in Li & Turner (2018) and popularized in causal discovery by Rolland et al. (2022); Montagna et al. (2023b); *(ii)* For

each environment $i \in [k]$, we need to find the observation $j \in [n]$ such that $\mathbf{f}^{-1}(\widehat{X}[i,j]) \approx \mu_{\mathbf{S}}$, that is, the data point generated mixing the source vector at the mean. Fortunately, this can be consistently estimated from the score $\nabla \log p_{\mathbf{x}}$, as we demonstrate in Proposition 2 in the appendix. These two steps are achieved by Algorithm 1 at the end of the first for loop. At this stage, all statistical quantities have been estimated: we note that, being the Stein estimator consistent, the algorithm is correct in the infinite sample limit. In the second for loop, we take the points at the estimated mean that we previously found, and compute the difference of the Hessians between the base and auxiliary environments: this exactly mirrors the first equality in Equations (10) and (11) of Lemma 1. Next, in the algorithm's notation, we compute

$$M := \widehat{H}_{\text{diffs}}^{-1}[1]\widehat{H}_{\text{diffs}}[2] \approx J_{\mathbf{f}}\Omega_1^{-1}\Omega_2 J_{\mathbf{f}^{-1}}. \tag{14}$$

Then, we solve the linear system $\widehat{H}_{\text{diffs}}[1]M = \widehat{H}_{\text{diffs}}[2]$ to find $M$. In the infinite samples limit Equation (14) is a precise equality, such that $M$ and $\Omega_1^{-1}\Omega_2$ are similar: diagonalizing $M$ we find $J_{\mathbf{f}^{-1}}$ up to a scaled permutation. The permutation indeterminacy is removed leveraging the assumption that the causal graph is acyclic via standard arguments (see Shimizu et al. (2006) and Reizinger et al. (2023)). Finally, the algorithm returns the estimated support of the inverse Jacobian.

**Analysis of the experiments.** In Figure 2 we illustrate the empirical performance of our method on several synthetic datasets generated from a bivariate causal model. We consider SCMs with the arbitrary nonlinear mechanisms *(i), (ii), (iii)* described in Section 4.1, and linear Gaussian models; as a sanity check, we also experiment on nonlinear additive noise models (ANM), post-nonlinear models (PNL), and location scale noise models (LSNM), which are all the nonlinear SCMs where identifiability can be achieved from observational data (see Section F.2 for details). All datasets are generated under the assumption that a causal effect exists (i.e., the ground truth graphs always have one arrow). We measure the errors through the structural hamming distance (SHD). This is equivalent to the number of edge additions, removals, or direction flips that are required to recover the ground truth graph from the estimated one: SHD=0 corresponds to correct inference, SHD=1 to an error. For each experimental configuration, consisting of function type and number of environments, we consider 50 seeds over which we compute the empirical mean and deviation of the SHD. The results are in line with our theory: we see that for the three models with *arbitrary* mechanisms, and the linear Gaussian SCM (all non-identifiable from pure observations), the average SHD is close to 0, which is especially evident when we do inference with only 3 environments. Interestingly, we see that adding environments doesn't always have a beneficial effect. This is not surprising, as we showed that two auxiliary environments are sufficient for inference. The method can also infer the causal direction for the ANM, PNL, and LSNM. We conclude that the empirical outcomes support our theory.

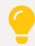 Our experiments validate the theoretical results: in the finite samples regime, when noise is Gaussian, only 2 sufficiently different auxiliary environments ere enough to identify causal directions.

*Remark on multivariate graphs.* Multivariate experiments are delayed to the Section F.5. On linear Gaussian SCMs, we find that our method can infer the causal order with only 3 environments for graphs up to 50 nodes, which is strong evidence in support of our theory. In the nonlinear setting, our method struggles to scale to high dimensions, and we limit our experiments to 5 nodes. A detailed discussion on the scalability of our approach is provided in the *Limitations* section B.2: in practice, scaling causal discovery with multiple environments beyond the bivariate setting is a well-known, unaddressed challenge, already found in Reizinger et al. (2023) and Monti et al. (2020). Given that algorithmic contributions fall beyond the scope of our paper, we leave this open problem for the future.

## 5 CONCLUSION

We demonstrated that the causal graph of a structural causal model with arbitrary nonlinear mechanisms is identifiable; surprisingly, this can be achieved given the auxiliary information of *only two* (sufficiently different) environments. Our main assumption is the Gaussianity of the noise terms, for which, however, we discuss potential relaxations. Our findings extend on the well-known duality between ICA and causal discovery: the first problem concerns the identifiability of the independent

sources at each point, whereas causality only needs to access the support of the Jacobian mixing function at *a single point*, when faithfulness is satisfied. The exciting consequence of this asymmetry is that while ICA identifiability requires a number of environments that grows linearly with the number of sources, for causal discovery, a constant number is sufficient: this makes our theoretical results appealing even in high dimensions. We hope that our work inspires novel identifiability theory beyond the Gaussianity constraint. Moreover, in light of our results, finding an efficient and effective algorithm for causal discovery with multiple environments and in high dimensions is a promising research direction.

**Reproducibility statement.** Section 4.1 describes the specifics for generating the synthetic data of our experiments. Section F.1 discusses the computational resources that were required for their execution. As supplementary material, we provide a zip folder that allows reproducing our empirical analysis. Particularly, it contains the Python code for: Algorithm 2, the synthetic data generation, the experiments execution, and the visualizations of the figures of this paper. For the theoretical results, we explicitly state and discuss in detail all the assumptions (Assumptions 1-5) required in Theorem 1 (our main contribution). A proof sketch and a detailed demonstration are included in the main text and the appendix, respectively (Section D.4).

## ACKNOWLEDGMENTS

FM is funded by the Chan Zuckerberg Initiative.

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

## CONTENTS

## A    LLM USAGE STATEMENT

In this work, LLMs were occasionally used for polishing and improving the writing. All research contributions in terms of theory and experiments' analysis were carried by the authors.

## B    LIMITATIONS

In this section, we discuss the limitations of our work and the open problems it leaves.

### B.1    THEORY

The main constraint in our theory is the requirement of Gaussian noise terms. In the main text (cf. Section 3.1, the paragraph *Theorem 1 beyond Gaussianity*), we discuss how this assumption is sufficient but might not be necessary. In fact, our theory can be extended to a structural causal model where the distribution of the sources has a vanishing gradient at some point. Our work does not address how to extend these result to arbitrary continuous distributions, which remains an open problem.

### B.2    EXPERIMENTS

#### B.2.1    SYNTHETIC DATA

One limitation in our work is that experiments are run on synthetic data. This is common in the causal discovery literature due to the challenge of accessing data with a reliable ground truth causal graph. Moreover, data collection often happens under the i.i.d. assumption: this hinders the application of our algorithm on common benchmarks such as, e.g., the Sachs dataset (Sachs et al., 2005), which doesn't dispose of multiple environments.

#### B.2.2    HIGH DIMENSIONAL GRAPHS

In Section F.5 we analyse experiments over graphs with more than 2 nodes. We find that, for linear Gaussian SCMs, our method can accurately infer the causal order of $50$ nodes with as few as three environments. However, for nonlinear structural causal models, performance quickly deteriorates with the number of dimensions. In general, we find that in the nonlinear setting, developing an effective algorithm for multivariate causal discovery with multiple environments is a challenging problem. This doesn't come as a surprise, being already well reported in the recent literature: Reizinger et al. (2023) (Table 1) show that for graphs with $5$ nodes, neural-based contrastive learning from multiple views fails to even converge to a causal order on $40\%$ of the test runs; on $10$ nodes, convergence occurs with a $27\%$ rate. Perhaps even more remarkable are the findings of Monti et al. (2020) (Figure 2) showing that, as the causal mechanisms become nonlinear, contrastive-based nonlinear ICA fails to recover the causal order better than a random baseline even for just two nodes. This highlights that algorithmic multi-environment causal discovery, even for small graphs, is an open and challenging problem that requires intensive research of its own–which is not in the scope of our paper.

Despite the clear limitation, it is important to keep in mind that the goal of our experiments is to demonstrate that the assumptions of Theorem 1–our main contribution–are sufficient to identify the causal direction, and not to present novel algorithmic contributions. To this end, bivariate models are well-known to be the easiest yet non-trivial setting: in fact, our experimental setup is reminiscent of that of Hoyer et al. (2008); Zhang & Hyvärinen (2009), two seminal papers in the identifiability theory of causality which limit their theoretical and empirical studies to bivariate causal graphs. This also aligns with several empirical and theoretical identifiability studies in causal discovery (e.g., Mooij et al. (2011); Ghassami et al. (2017); Montagna et al. (2024); Immer et al. (2022); Xi et al. (2025); Monti et al. (2020); Strobl & Lasko (2023)), which makes our choice to focus on two-variable graphs well-justified. We leave the challenge of developing an algorithm suitable for multi-environment causal discovery in higher dimensions as an open problem.

## C   RELATED WORKS

**Soft interventions and multiple environments for causal discovery.**   Several works in the literature have addressed causal discovery identifiability and estimation via non i.i.d. data (stemming from soft interventions and multiple environments). Peters et al. (2015) and Heinze-Deml et al. (2018) identify the parents of a designated target node via invariance across environments, yielding partial identifiability of causal directions. They assume linear and nonlinear additive noise models, respectively. Huang et al. (2020) use nonstationarity to recover the skeleton and orient some edges. Perry et al. (2022) leverage sparse mechanism shifts, proving high-probability graph recovery with bounds that improve as the number of environments grows. Rothenhäusler et al. (2015) is the closest to our work, but their results are limited to linear models. Ghassami et al. (2017; 2018) and Heurtebise et al. (2025), similarly to our work, study identifiability of structural causal models from multiple environments, but their identifiability results are specialized to the linear case. Recently, Jalaldoust et al. (2025) formulated a statistical test that can find a superset of the parents of a target node. Yang et al. (2018); Brouillard et al. (2020); Jaber et al. (2020) characterize equivalence classes identifiability from interventions. From a methodological perspective, Brouillard et al. (2020); Ke et al. (2023) introduce differentiable approaches to causal discovery with interventions; Mooij et al. (2020) propose a unifying framework for causal discovery from observational and multi-environment data. All of these results are complementary to our work, which is, to the best of our knowledge, the first to provide guarantees of identifiability of the causal graph from a finite number of auxiliary additional environments, potentially only two.

**ICA and causal discovery.**   The seminal work of Shimizu et al. (2006) shows that if an SCM can be expressed as a linear non-Gaussian ICA model, the underlying causal graph is identifiable. Reizinger et al. (2023) generalize this to the nonlinear case. Monti et al. (2020) show that time contrastive ICA (Hyvärinen & Morioka, 2016) can identify bivariate causal graphs with arbitrary nonlinear mechanism. The common ground of these findings is that they adapt the existing ICA identifiability theory to the problem of causal discovery. This approach is clearly important, especially in the light of the recent advancement in multi-environment ICA identifiability (Hyvärinen et al., 2019; Khemakhem et al., 2020a;b; Gresele et al., 2019; Hälvä & Hyvärinen, 2020; Hyvärinen & Morioka, 2017; Hälvä et al., 2021); however, in the nonlinear setting, it fails to capture the gap between the two problems: while ICA attempts to recover the mixing function and the independent sources at each point, causal discovery concerns the much simpler problem of structure identifiability. Our work shows that this difference is key to demonstrating causal discovery identifiability from a constant number of sufficiently different environments, where ICA requires at least as many as the number of sources (see e.g. Theorem 1 in Hyvärinen & Morioka (2016)).

## D   PROOF OF THE THEORETICAL RESULTS

### D.1   PRELIMINARY THEORETICAL RESULTS

In this section, we collect the theoretical results useful for the proof of Theorem 1.

**Lemma 2** (Full rank of $\Omega_l$ under rescalings). *Assume Gaussian sources* $\mathbf{S}$ *with independent coordinates, and environments generated by rescalings* $\mathbf{S}^e = L_e \mathbf{S}$ *with* $L_e = \mathrm{diag}(\lambda_1^e, \ldots, \lambda_d^e)$ *and* $\lambda_j^e \neq 0$. *For* $l \in \{1, 2\}$, *recall the index sets* $\mathcal{E}_1$ *and* $\mathcal{E}_2$, *and*

$$\Omega_l := \sum_{e \in \mathcal{E}_l} \left( D_s^2 \log p_\theta(\mathbf{s}) - D_s^2 \log p_\theta^e(\mathbf{s}) \right),$$

*evaluated at the same* $\mathbf{s}$. *Then each* $\Omega_l$ *is diagonal with entries*

$$(\Omega_l)_{jj} = \frac{1}{\sigma_j^2} \left( \sum_{e \in \mathcal{E}_l} \frac{1}{(\lambda_j^e)^2} - |\mathcal{E}_l| \right),$$

*and therefore*

$$\Omega_l \text{ is full rank} \iff \forall j \in [d] : \sum_{e \in \mathcal{E}_l} \frac{1}{(\lambda_j^e)^2} \neq |\mathcal{E}_l|.$$

*Proof.* For a univariate Gaussian, $D_{s_j}^2 \log p(s_j) = -1/\sigma_j^2$. In environment $e$ we have $S_j^e = \lambda_j^e S_j$, so $S_j^e$ has variance $(\lambda_j^e \sigma_j)^2$, hence $D_{s_j}^2 \log p^e(s_j) = -1/(\lambda_j^e \sigma_j)^2$. Thus

$$\left(D_{s_j}^2 \log p(s_j) - D_{s_j}^2 \log p^e(s_j)\right) = \frac{1}{\sigma_j^2}\left(\frac{1}{(\lambda_j^e)^2} - 1\right).$$

Summing over $e \in \mathcal{E}_l$ gives the stated diagonal form. A diagonal matrix is full rank iff none of its diagonal entries is zero, which yields the equivalence. $\square$

**Lemma 3.** $\Omega_l$ *is invertible implies* $\widehat{\Omega}_l$ *invertible.*

*Proof.* By Lemma 1, for $l = 1, 2$, we have:

$$J_{\mathbf{f}^{-1}}(\mathbf{x})^T \Omega_l J_{\mathbf{f}^{-1}}(\mathbf{x}) = J_{\widehat{\mathbf{f}}^{-1}}(\mathbf{x})^T \widehat{\Omega}_l J_{\widehat{\mathbf{f}}^{-1}}(\mathbf{x}).$$

Under Assumption 5, by Lemma 2 the LHS is a product of full rank matrices, and so is full rank; so must be the RHS. Given that $\operatorname{rank}(AB) \le \min\left(\operatorname{rank}(A), \operatorname{rank}(B)\right)$ (for generic matrices $A, B$) we conclude that $\widehat{\Omega}_l$ is also full rank. $\square$

## D.2 PROOF OF LEMMA 1

We report the content of Lemma 1, followed by its proof.

**Lemma 1.** *Let* $\mathbf{x} = \mathbf{f}(\mathbf{s}) = \widehat{\mathbf{f}}(\hat{\mathbf{s}})$, *where* $\mathbf{s} = \mu_{\mathbf{S}}$. *Let Assumptions 1,2 and 4 satisfied. Then:*

$$\sum_{e \in \mathcal{E}_1} D_{\mathbf{x}}^2 \log p(\mathbf{x}) - D_{\mathbf{x}}^2 \log p^e(\mathbf{x}) = J_{\mathbf{f}^{-1}}(\mathbf{x})^T \Omega_1 J_{\mathbf{f}^{-1}}(\mathbf{x}) = J_{\widehat{\mathbf{f}}^{-1}}(\mathbf{x})^T \widehat{\Omega}_1 J_{\widehat{\mathbf{f}}^{-1}}(\mathbf{x}) \quad (10)$$

$$\sum_{e \in \mathcal{E}_2} D_{\mathbf{x}}^2 \log p(\mathbf{x}) - D_{\mathbf{x}}^2 \log p^e(\mathbf{x}) = J_{\mathbf{f}^{-1}}(\mathbf{x})^T \Omega_2 J_{\mathbf{f}^{-1}}(\mathbf{x}) = J_{\widehat{\mathbf{f}}^{-1}}(\mathbf{x})^T \widehat{\Omega}_2 J_{\widehat{\mathbf{f}}^{-1}}(\mathbf{x}) \quad (11)$$

*Proof.* By direct computation, it can be verified that for each $e \in \{0\} \cup \mathcal{E}$, we have:

$$D_{\mathbf{x}}^2 \log p^e(\mathbf{x}) = D_{\mathbf{x}}^2 \log |J_{\mathbf{f}^{-1}}(\mathbf{x})| + J_{\mathbf{f}^{-1}}(\mathbf{x})^T D_{\mathbf{s}}^2 \log p_\theta^e(\mathbf{s}) J_{\mathbf{f}^{-1}}(\mathbf{x})$$
$$+ \sum_{k=1}^d \partial_{s_k} \log p_\theta^e(s_k) D_{\mathbf{x}}^2 \mathbf{f}_k^{-1}(\mathbf{x}). \quad (15)$$

Given $\mathbf{s} = \mu_{\mathbf{S}}$, Assumption 4 of Gaussianity, together with the fact that $\mathbf{S}^e$ is distributionally equivalent to $L_e \mathbf{S}$ for some diagonal $L_e$ (Assumption 2), imply $\partial_{s_k} \log p_\theta^e(s_k) = 0$ for all $k$. Then, the summation vanishes. It follows that, for all environments $e \in \mathcal{E}$:

$$D_{\mathbf{x}}^2 \log p(\mathbf{x}) - D_{\mathbf{x}}^2 \log p^e(\mathbf{x}) = J_{\mathbf{f}^{-1}}(\mathbf{x})^T \left(D_{\mathbf{s}}^2 \log p_\theta(\mathbf{s}) - D_{\mathbf{s}}^2 \log p_\theta^e(\mathbf{s})\right) J_{\mathbf{f}^{-1}}(\mathbf{x}).$$

The same results hold if we replace $\mathbf{f}$ with $\widehat{\mathbf{f}}$ and $\theta$ with $\hat{\theta}$. Then, Equation (10) follows summing the above over all $e \in \mathcal{E}_1$, and Equation (11) follows summing over $e \in \mathcal{E}_2$. $\square$

## D.3 IDENTIFIABILITY OF THE MEAN OF THE SOURCES

In this section, we show that under the assumptions of Theorem 1, the mean $\mu_{\mathbf{S}}$ of the sources is identifiable.

**Proposition 2** (Identifiability of the sources mean)**.** *For each* $e \in \mathcal{E}$, *suppose the diagonal entries of the rescaling matrices* $L_e$ *generating the environments are randomly drawn from a joint distribution that is absolutely continuous with respect to the Lebesgue measure on* $(\mathbb{R} \setminus 0)^{d|\mathcal{E}|}$. *Then, the following is verified with probability one over the samples* $\{L_e\}_{e \in \mathcal{E}}$:

$$\sum_{e=1}^k \nabla \log p(\mathbf{x}) - \nabla \log p^e(\mathbf{x}) = 0 \iff \mathbf{s} = \mathbf{f}^{-1}(\mathbf{x}) = \mu_{\mathbf{S}}. \quad (16)$$

We introduce two lemmas instrumental to the proof of the proposition.

**Lemma 4.** *Consider the base ICA model of Equation* (2), *and let* $e = 1, ..., k$ *be the index denoting an auxiliary environment (cf. Equation* (4)). *Let Assumptions* 1-4 *to be satisfied. Given* $\mathbf{x} = \mathbf{f}(\mathbf{s})$ *such that* $J_{\mathbf{f}^{-1}}(\mathbf{x})$ *is full rank, for each* $k \leq |\mathcal{E}|$:

$$\sum_{e=1}^{k} \nabla \log p(\mathbf{x}) - \nabla \log p^e(\mathbf{x}) = 0 \iff \sum_{e=1}^{k} \nabla \log p(\mathbf{s}) - \nabla \log p^e(\mathbf{s}) = 0 \qquad (17)$$

*Proof.* By the change of variable formula for densities, we obtain the score of $\mathbf{x}$ for a generic environment $e = 0, ..., k$ (as usual, $p = p^0$):

$$\nabla \log p^e(\mathbf{x}) = J_{\mathbf{f}^{-1}}(\mathbf{x})^T \nabla \log p^e(\mathbf{s}) + \nabla \log |J_{\mathbf{f}^{-1}}(\mathbf{x})|.$$

Then, for each $e = 1, ..., k$:

$$\nabla \log p(\mathbf{x}) - \nabla \log p^e(\mathbf{x}) = J_{\mathbf{f}^{-1}}(\mathbf{x})^T \left[ \nabla \log p(\mathbf{s}) - \nabla \log p^e(\mathbf{s}) \right].$$

Taking the summation:

$$\sum_{e=1}^{k} \nabla \log p(\mathbf{x}) - \nabla \log p^e(\mathbf{x}) = \sum_{e=1}^{k} J_{\mathbf{f}^{-1}}(\mathbf{x})^T \left[ \nabla \log p(\mathbf{s}) - \nabla \log p^e(\mathbf{s}) \right].$$

From the above equation, the right-to-left implication trivially holds. Considering the other direction we have:

$$\sum_{e=1}^{k} \nabla \log p(\mathbf{x}) - \nabla \log p^e(\mathbf{x}) = 0 \implies \sum_{e=1}^{k} J_{\mathbf{f}^{-1}}(\mathbf{x})^T \left[ \nabla \log p(\mathbf{s}) - \nabla \log p^e(\mathbf{s}) \right] = 0.$$

Being the Jacobian of the inverse mixing function a full rank matrix, its null space is the zero vector, which implies:

$$\sum_{e=1}^{k} \nabla \log p(\mathbf{s}) - \nabla \log p^e(\mathbf{s}) = 0.$$

$\square$

**Lemma 5.** *Consider the base ICA model* $\mathbf{X} = \mathbf{f}(\mathbf{S})$ *of Equation* (2). *Let* $e = 1, ..., k$ *be the index of the auxiliary environment* $\mathbf{X}^e = \mathbf{f}(\mathbf{S}^e)$, *with* $\mathbf{S}^e = L_e \mathbf{S}$, $L_e = \text{diag}(\lambda_1^e, \dots, \lambda_d^e)$, *and* $\lambda_j^e \neq 0$. *Let Assumptions* 1 *and* 4 *be satisfied. Assume the joint law of* $\{\lambda_j^e : j = 1, \dots, d, \ e = 1, \dots, k\}$ *is absolutely continuous with respect to Lebesgue measure on* $(\mathbb{R} \setminus \{0\})^{dk}$. *Then, for each* $k \leq |\mathcal{E}|$, *the following holds with probability one over* $\{L_e\}_{e=1}^k$ *samples:*

$$\sum_{e=1}^{k} \nabla \log p(\mathbf{s}) - \nabla \log p^e(\mathbf{s}) = 0 \iff \mathbf{s} = \mathbf{f}^{-1}(\mathbf{x}) = \mu_{\mathbf{S}}. \qquad (18)$$

*Proof.* The backward direction is immediate, due to the Gaussianity assumption. Let's focus on the forward implication.

$$\sum_{e=1}^{k} \nabla \log p(\mathbf{s}) - \nabla \log p^e(\mathbf{s}) = 0 \iff \sum_{e=1}^{k} \partial_{s_j} \log p(s_j) - \partial_{s_j} \log p^e(s_j) = 0, \ \forall j = 1, ..., d.$$

We denote with $\mu_j, \sigma_j^2$ respectively the mean and variance of $S_j$, and define $\lambda_j^0 := 1$. For each $e = 0, ..., k$ we have:

$$\partial_{s_j} \log p^e(s_j) = \frac{\mu_j - s_j}{(\lambda_j^e \sigma_j)^2}.$$

Then:

$$\sum_{e=1}^{k} \partial_{s_j} \log p(s_j) - \partial_{s_j} \log p^e(s_j) = \frac{\mu_j - s_j}{\sigma_j^2} \left( k - \sum_{e=1}^{k} \frac{1}{(\lambda_j^e)^2} \right).$$

Therefore, the sum vanishes if and only if for every $j$, either $s_j = \mu_j$ or $\sum_{e=1}^{k} (\lambda_j^e)^{-2} = k$. By Proposition 3, $\sum_{e=1}^{k} (\lambda_j^e)^{-2} = k$ occurs with probability zero, and thus the claim is verified.

$\square$

We are ready to prove the proposition.

*Proof of Proposition 2.* By Lemma 4 we have that for each $k \leq |\mathcal{E}|$:

$$\sum_{e=1}^{k} \nabla \log p(\mathbf{x}) - \nabla \log p^e(\mathbf{x}) = 0 \iff \sum_{e=1}^{k} \nabla \log p(\mathbf{s}) - \nabla \log p^e(\mathbf{s}) = 0$$

Then, the result follows by application of Lemma 5. □

### D.4 PROOF OF THEOREM 1

We repropose the statement of Theorem 1, followed by a detailed proof.

**Theorem 1.** *Consider the groundtruth ICA model $(\mathbf{f}, p_\theta)$ of Equation (2) and the alternative $(\widehat{\mathbf{f}}, p_{\hat\theta})$. Let Assumptions 1-5 be satisfied, and assume that the elements in the set $\{(\Omega_1^{-1}\Omega_2)_{ii}\}_{i=1}^{d}$ are pairwise distinct. Let $\mathbf{x} = \mathbf{f}(\mathbf{s}) = \widehat{\mathbf{f}}(\hat{\mathbf{s}})$ and $\mathbf{s} = \mu_\mathbf{S}$: then, the indeterminacy function $\mathbf{h} := \widehat{\mathbf{f}}^{-1} \circ \mathbf{f}$ satisfies $J_\mathbf{h}(\mathbf{s})$ full rank and diagonal, meaning that the causal graph $\mathcal{G}$ is identifiable.*

*Proof.* By Lemma 1, for $l = 1, 2$ we have:

$$J_{\mathbf{f}^{-1}}(\mathbf{x})^T \Omega_l J_{\mathbf{f}^{-1}}(\mathbf{x}) = J_{\widehat{\mathbf{f}}^{-1}}(\mathbf{x})^T \widehat{\Omega}_l J_{\widehat{\mathbf{f}}^{-1}}(\mathbf{x}),$$

which implies

$$M^T \Omega_l M = \widehat{\Omega}_l, \quad l = 1, 2, \tag{19}$$

where $M := J_{\mathbf{h}^{-1}}(\hat{\mathbf{s}})$. By Lemma 2, $\Omega_l$ is invertible, which also implies $\widehat{\Omega}_l$ invertibility (by Lemma 3). Then, we can define $A := \widehat{\Omega}_1^{-1}\widehat{\Omega}_2$ and $B := \Omega_1^{-1}\Omega_2$. From Equation (19) it follows:

$$A = M^{-1} B M, \tag{20}$$

which implies that $A$ and $B$ are similar, implying that they have the same set of eigenvalues. Take $\lambda, \mathbf{v}$ eigenvectors of $A$. Then, the following chain of implication holds:

$$A\mathbf{v} = \lambda\mathbf{v} \iff MA\mathbf{v} = \lambda M\mathbf{v} \iff BM\mathbf{v} = \lambda M\mathbf{v}, \tag{21}$$

where the last step follows from Equation (20). So, $M$ is mapping from eigenvectors of $A$ to eigenvectors of $B$. The next step is showing that each eigenspace of $A$ and $B$ is always spanned by one vector in the standard basis. As a preliminary step, we show that the diagonal elements of $A$ are pairwise distinct: first, by similarity, we have that $A$ and $B$ have the same eigenvalues. Being both matrices diagonal, the eigenvalues are the diagonal elements. Then:

$$A_{ii} = \frac{(\widehat{\Omega}_1)_{ii}}{(\widehat{\Omega}_2)_{ii}} = \frac{(\Omega_1)_{jj}}{(\Omega_2)_{jj}} = B_{jj}, \quad i, j = 1, ..., d. \tag{22}$$

By assumption, we have that the elements in the set $\{\frac{(\Omega_1)_{\ell\ell}}{(\Omega_2)_{\ell\ell}}\}_{\ell \in [d]}$ are pairwise distinct. The above equation implies the same for the set $\{\frac{(\widehat{\Omega}_1)_{\ell\ell}}{(\widehat{\Omega}_2)_{\ell\ell}}\}_{\ell \in [d]}$, i.e., for each $i = 1, ..., d$:

$$A_{ii} \neq A_{jj}, \quad \forall j = 1, ..., d, j \neq i. \tag{23}$$

Now consider the eigenvalue $\lambda$ of $A$: we show that the associated eigenspace is equal to the span of a single vector in the standard basis. Being $A$ diagonal, there is $i = 1, ..., d$ such that $\lambda = A_{ii}$. Consider the eigenvector $\mathbf{v} = (v_1, ..., v_d)$ such that:

$$A\mathbf{v} = \lambda\mathbf{v} = A_{ii}\mathbf{v}. \tag{24}$$

Being $A$ diagonal, for each $j = 1, ..., d$, component-wise we have:

$$(A\mathbf{v})_j = A_{jj}v_j. \tag{25}$$

Equations (24) and (25) together imply:

$$A_{ii}v_j = A_{jj}v_j \iff (A_{ii} - A_{jj})v_j = 0, \quad \forall j = 1, ..., d.$$

By Equation (23), for $i \neq j$, $A_{ii} \neq A_{jj}$, meaning that $v_j = 0$. Then, $\mathbf{v}$ eigenvector of $A$ must be aligned with the basis vector $\mathbf{e}_i$:

$$E_\lambda(A) = \mathrm{span}\{\mathbf{e}_i\}. \tag{26}$$

With analogous computations, we find:

$$E_\lambda(B) = \mathrm{span}\{\mathbf{e}_j\}, \tag{27}$$

with $\mathbf{e}_j$ potentially different from $\mathbf{e}_i$. Given that by Equation (21) we have $M E_\lambda(A) = E_\lambda(B)$, the last two equations imply

$$M \,\mathrm{span}\{\mathbf{e}_i\} = \mathrm{span}\{\mathbf{e}_j\}, \quad M = J_\mathbf{h}(\mathbf{s}).$$

We conclude that $J_\mathbf{h}(\mathbf{s})$ maps one vector in the standard basis to another (up to rescaling), proving that $J_\mathbf{h}(\mathbf{s}) = DP$ with $D$ invertible diagonal and $P$ permutation. We recall that by Equation (8) we have $J_\mathbf{f} = J_{\widehat{\mathbf{f}}} J_\mathbf{h}$, s.t.

$$J_{\mathbf{f}^{-1}}(\mathbf{x}) = P^T D^{-1} J_{\widehat{\mathbf{f}}^{-1}}(\mathbf{x}).$$

By Lemma 1 in Reizinger et al. (2023), the permutation indeterminacy can be uniquely determined and thus removed. Given that by Assumption 3 the Jacobian of $J_{\mathbf{f}^{-1}}(\mathbf{x})$ is faithful to the causal graph, the claim is verified. $\quad\square$

# E  INDEPENDENT COMPONENT ANALYSIS

In this section, we present a primer on the problem of Independent Component Analysis (ICA), based on the content of Section 2 in Buchholz et al. (2022). ICA seeks to recover latent *sources* from their observed mixtures. We assume a hidden random vector $\mathbf{S} \in \mathbb{R}^d$ with independent coordinates and observations generated by

$$\mathbf{X} = \mathbf{f}(\mathbf{S}), \qquad p(\mathbf{s}) = \prod_{i=1}^d p_i(s_i), \tag{28}$$

where $\mathbf{f} : \mathbb{R}^d \to \mathbb{R}^d$ is a diffeomorphism. The goal of ICA is to find an *unmixing* map $\widehat{\mathbf{f}}^{-1} : \mathbb{R}^d \to \mathbb{R}^d$ such that the components of $\widehat{\mathbf{f}}^{-1}(\mathbf{X})$ are independent—ideally achieving blind source separation (BSS), meaning $\widehat{\mathbf{f}}^{-1} \approx \mathbf{f}^{-1}$ up to standard symmetries. Informally, for $\hat{\mathbf{s}} = \widehat{\mathbf{f}}^{-1}(\mathbf{x})$, we call $\widehat{\mathbf{f}}$ an ICA solution when

$$\widehat{\mathbf{f}}(\hat{\mathbf{s}}) \overset{D}{=} \mathbf{f}(\mathbf{s})$$

(equality is in distribution). In general, we would like an ICA solution to be as close as possible to the real function $\mathbf{f}$. To formalize this concept, known as *identifiability*, let $\mathcal{F}(\mathcal{A}, \mathcal{B})$ be a class of invertible maps $\mathcal{A} \to \mathcal{B}$ (assumed diffeomorphisms) and let $\mathcal{P} \subset \mathcal{M}_1(\mathbb{R})^{\otimes d}$ be a family of product measures. Let $\mathcal{S}$ denote the group of admissible *symmetries* (e.g., permutations and coordinate-wise rescalings) up to which we agree to identify sources.

**Definition 3** (Identifiability). *ICA in $(\mathcal{F}, \mathcal{P})$ is identifiable up to $\mathcal{S}$ if, for any $\mathbf{f}, \widehat{\mathbf{f}} \in \mathcal{F}$ and $P, \hat{P} \in \mathcal{P}$,*

$$\mathbf{f}(\mathbf{S}) \overset{D}{=} \widehat{\mathbf{f}}(\hat{\mathbf{S}}) \quad \textit{with } \mathbf{S} \sim P, \ \hat{\mathbf{S}} \sim \hat{P}, \tag{29}$$

*implies the existence of $\mathbf{h} \in \mathcal{S}$ such that $\mathbf{h} = \widehat{\mathbf{f}}^{-1} \circ \mathbf{f}$ on the support of $P$.*

In general (i.e., for $(\mathcal{F}, \mathcal{P})$ arbitrarily large), the ICA problem is not identifiable for reasonable $\mathcal{S}$. Notable example comes from the Darmois construction or constructions based on measure-preserving transformations. Several results in the literature have studied which conditions on $(\mathcal{F}, \mathcal{P})$ can help identifiability. Most notably, Buchholz et al. (2022) shows that when $\mathcal{F}$ represents the class of conformal maps, identifiability is guaranteed up to trivial indeterminacies. If heterogeneous data are considered (e.g., in the multi-environment setting of this paper), identifiability was shown in the general case (Hyvärinen & Morioka, 2016).

# F  EXPERIMENTS APPENDIX

## F.1  COMPUTATIONAL RESOURCES

All experiments have been run on a personal laptop, a Lenovo ThinkPad T14 Gen 5, for a run time of approximately 6 hours.

## F.2 Structural causal model identifiability from observational data

Without sufficiently restrictive modeling assumptions, causal discovery is ill-posed: the distribution of the data is compatible with many distinct graphs that define an equivalence class, the most one can hope to identify in the general case with i.i.d. observations. Unique graph recovery requires restrictions on the class of functional mechanisms and noise distributions of the underlying causal model: in what follows, we briefly introduce the four classes of causal models that are known to be identifiable. We always assume that the underlying graph is a DAG.

**Linear Non-Gaussian Model (LiNGAM).** A linear SCM over $\mathbf{X} \in \mathbb{R}^d$ is defined by

$$\mathbf{X} = B\mathbf{X} + \mathbf{S}, \tag{30}$$

where $B \in \mathbb{R}^{d \times d}$ collects the coefficients expressing each $X_i$ as a linear function of its parents plus a disturbance $S_i$. With mutually independent, non-Gaussian noise terms, the model is identifiable; this is known as the Linear Non-Gaussian Acyclic Model (LiNGAM) (Shimizu et al., 2006).

**Additive Noise Model (ANM).** An Additive Noise Model (ANM) (Hoyer et al., 2008; Peters et al., 2014) defines each causal variable as a function of (potentially) nonlinear mechanisms and an additive noise contribution:

$$X_i := f_i(\mathrm{PA}_i) + S_i, \quad i = 1, \ldots, d. \tag{31}$$

The noise terms are required to be mutually independent.

**Post-Nonlinear Model (PNL).** The most general class with known sufficient conditions for identifiability of the graph is the Post-Nonlinear (PNL) model (Zhang & Hyvärinen, 2009), in which

$$X_i := g_i\big(f_i(\mathrm{PA}_i) + S_i\big), \quad i = 1, \ldots, d, \tag{32}$$

with $f_i$ and $g_i$ both potentially nonlinear, $g_i$ invertible, and mutually independent noises.

**Location Scale Noise Model (LSNM)** The LSNM (Immer et al., 2022) extends ANMs by allowing heteroscedastic noise as follows:

$$X_i := f_i(\mathrm{PA}_i) + g_i(\mathrm{PA}_i)\,S_i, \quad i = 1, \ldots, d, \tag{33}$$

where $f_i$ and $g_i > 0$ may be nonlinear and noise terms are jointly independent with zero mean and unit variance.

## F.3 Detailed pseudocode of Algorithm 1

Algorithm 2 provides a detailed pseudocode of the algorithm adopted in our experiments of Section 4, and sketched in Algorithm 1.

## F.4 Experiments beyond Gaussianity

In this section, we present additional experimental results on bivariate graphs underlying synthetically generated structural causal models. The causal mechanisms are the same already described in Section 4.1. The difference, here, is that we generate the independent sources from a Gamma distribution, which violates the assumptions of our theory. We sample the scale parameter $\theta \sim U(1.75, 2.25)$, and consider two different parameterizations of the shape $\alpha$ of the base environments: in the first case, $\alpha \sim U(0.5, 1)$; in the second case $\alpha \sim U(2, 2.5)$. What makes the Gamma density interesting it that it can be flexibly modified by changing the values of its parameters, as shown in Figure 3.

**Gamma distribution with no vanishing gradient.** Figure 3a illustrate how the Gamma density function varies at $\alpha = 1$ and different values of $\theta$. It is interesting to note that the gradient of the density function never vanishes, making this setup adversarial to the assumptions of Theorem 1. In line with this, in Figure 4 we see that generally our algorithm struggles to infer the causal direction for this class of structural causal models.

---

**Algorithm 2:** Estimating supp $J_{\mathbf{f}^{-1}}$ from the data

---

**Data:** $\mathcal{D} \in \mathbb{R}^{k \times n \times d}$      `// ∀ env: n d-dimensional observations.`

    $\mathcal{E}_1, \mathcal{E}_2 \subset [k]$      `// Set of indices splitting the environments in two groups`

**Result:** Estimate of supp $J_{\mathbf{f}^{-1}}$

$\widehat{S} \leftarrow \text{score\_estimate}(\mathcal{D}) \in \mathbb{R}^{k \times n \times d}$

$\widehat{H} \leftarrow \text{hess\_estimate}(\mathcal{D}) \in \mathbb{R}^{k \times n \times d \times d}$

$\text{mean\_pairs\_idxs} \in \mathbb{R}^{k \times 2}$      `// Pair of indices corresponding to observations at the mean`

`// For each environment e, find i s.t.` $\mathbf{f}^{-1}(X[e,i]) \approx \mu_{\mathbf{S}}$

**for** $e = 1, ..., k$ **do**

    $\Delta_X \in \mathbb{R}^{n \times n}$      `// norm of the difference of observations from distinct envs`

    $\text{pairs} \in \mathbb{N}^n$      `// Pair of indices i,j such that` $X[0,i] \approx X[e,j]$

    $\text{score\_diffs} \leftarrow +\infty \in \mathbb{R}^n$      `// Container for norm of the differences in the score`

    **for** $i = 1,...,n$ **do**

        **for** $j=1,...,n$ **do**

            $\Delta_X[i,j] \leftarrow ||X[0,i] - X[e,j]||_2$

        **end**

        $j \leftarrow \arg\min \Delta_X[i]$

        $\text{pairs}[i] \leftarrow j$      `//` $X[0,i] \approx X[e,j]$

        $\text{score\_diffs}[i] \leftarrow ||\widehat{S}[0,i] - \widehat{S}[e,j]||_2$

    **end**

    $m \leftarrow \arg\min \text{score\_diffs}$      `// Paired observations between envs (0,e) s.t. score diff. ≈0.`

    $\text{mean\_pairs\_idxs}[e] \leftarrow m, \text{pairs}[m]$      `// The score diff. vanishes when source = mean`

**end**

`// Difference of Hessians at the mean (i.e. Equations (10) and (11))`

$\widehat{H}_{\text{diffs}} \leftarrow 0 \in \mathbb{R}^{2 \times d \times d}$

**for** $\ell = 1, 2$ **do**

    **for** $e \in \mathcal{E}_\ell$ **do**

        $m_1, m_e \leftarrow \text{mean\_pairs\_idxs}[e]$

        $\Delta_H = \widehat{H}[0, m_1] - \widehat{H}[e, m_e]$

        $\widehat{H}_{\text{diffs}}[\ell] \leftarrow \widehat{H}_{\text{diffs}}[\ell] + \Delta_H.$

    **end**

**end**

$M \leftarrow \widehat{H}_{\text{diffs}}^{-1}[1]\widehat{H}_{\text{diffs}}[2] \approx J_{\mathbf{f}}\Omega_1^{-1}\Omega_2 J_{\mathbf{f}^{-1}}$      `//` $H_{\text{diffs}}[\ell] \approx J_{\mathbf{f}^{-1}}^T \Omega_\ell J_{\mathbf{f}^{-1}}$`, by Equations (10) and (11)`

$\widehat{J}_{\mathbf{f}^{-1}} \leftarrow \text{diagonalize}(M) \approx J_{\mathbf{f}^{-1}}DP$

**return** supp $\left(\widehat{J}_{\mathbf{f}^{-1}}P^{-1}\right)$      `// P can be found using the acyclicity of the causal graph.`

---

**Gamma distribution with vanishing gradient.** Figure 3b illustrates how the Gamma density function varies at $\alpha = 2$ and different values of $\theta$. We can see that, in this case, the density achieves a maximum: we point to our analysis in Section 3.1 (the paragraph *Theorem 1 beyond Gaussianity*), where we discuss when and why it is reasonable to expect that Theorem 1 extends to any source distribution that achieves a maximum or minimum in the interior of its domain. A word of caution is needed: despite the fact that the Gamma density with $\alpha \in [2, 2.5]$ does have a vanishing gradient, the points of the domain at which the critical values occur are not preserved by our rescaling interventions (as is clear by inspection of Figure 3b). Hence, the requirements of the Theorem 1 are not fully met (where it's implicit that the rescaling interventions do not change the location of the modes): this makes the experiments of Figure 5 an interesting challenge for our algorithm. The outcomes are exciting: we see that increasing the number of available environments, despite the assumption violations, imposes enough constraints to infer the causal direction in the majority of the experimental setups with $\approx 80\%$ accuracy. This is of double interest: first, we have some empirical

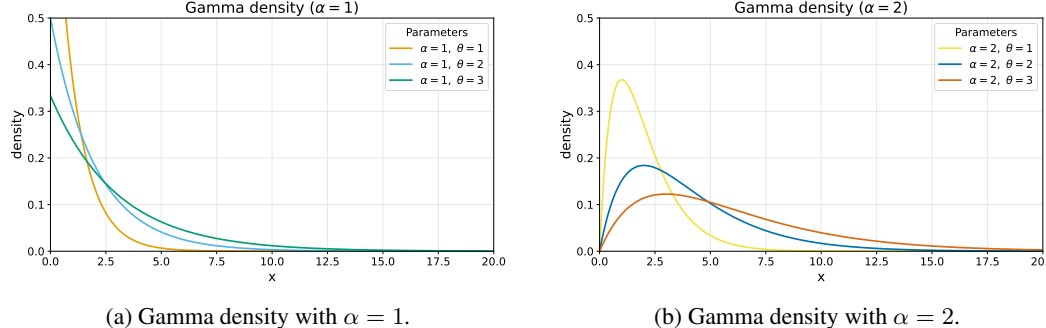

(a) Gamma density with $\alpha = 1$.

(b) Gamma density with $\alpha = 2$.

Figure 3: We plot the Gamma density for different values of shape and scale. The left plot fixes the shape $\alpha = 1$; the right plot fixes $\alpha = 2$. We let $\theta$ vary to illustrate how the distribution changes between the rescaling environments of our experiments. We note that for $\alpha = 1$ the density doesn't have a finite critical point.

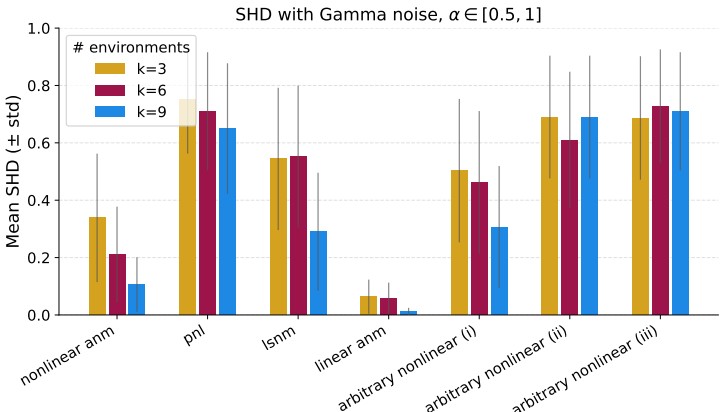

Figure 4: Average SHD ($0$ is best, $1$ is worst) achieved by Algorithm 1 over $50$ seeds on binary graphs. The sources are sampled from a gamma distribution with $\alpha \in [0.5, 1]$. In line with our theory, when the sources are generated according to a density that doesn't have critical points, our algorithm generally fails to infer the causal direction.

evidence supporting the hypothesis that our theory can be extended beyond Gaussianity. Second, we see that this seems to be achieved thanks to the constraints from many environments, in contrast with what we observe when experiments are run on SCMs with Gaussian noise (Figure 2), where increasing environments do not translate into better accuracy. These empirical findings, despite being preliminary, should provide an incentive to pursue identifiability theory beyond Gaussianity.

## F.5    EXPERIMENTS ON HIGHER DIMENSIONAL GRAPHS

In this section, we present and analyse experimental results on graphs in dimensions higher than 2. Our finding shows that, according to our theory, 2 sufficiently different auxiliary environments are enough to infer about the causal order, even in cases known to be non-identifiable with pure observations.

**Metric.**   We monitor the error in the inferred causal order via the topological order divergence, first adopted in Rolland et al. (2022). Given a directed acyclic graph with $d$ nodes, a causal order (or *topological* order) is a permutation of the set $[d]$ such that a node in the ordering can be a parent only of the nodes appearing after it in the same ordering. For example, the only graphs compatible with the topological order $\{2, 1\}$ are $X_2 \rightarrow X_1$ or the empty graph. Consider a causal order $\hat{\pi}$, and

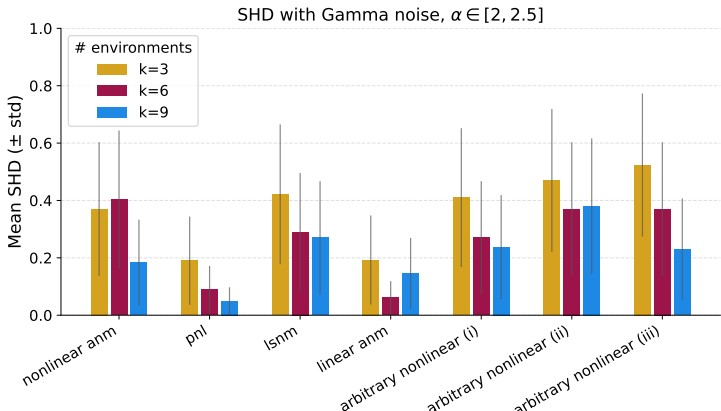

Figure 5: Average SHD (0 is best, 1 is worst) achieved by Algorithm 1 over 50 seeds on binary graphs. The sources are sampled from a gamma distribution with $\alpha \in [2, 2.5]$, which guarantees at least one point where the gradient of the log-likelihood vanishes (see Figure 3b). Interestingly, this appears to enable accurate inference of the causal graph when the number of environments increases.

a binary adjacency matrix $A$ representing a directed acyclic graph ($A_{ij} = 1 \iff i \in \mathrm{PA}_j$). The topological order divergence is defined as:

$$D_{\mathrm{top}}(\hat{\pi}, A) = \sum_{i=1}^{d} \sum_{j:\hat{\pi}_i > \hat{\pi}_j} A_{ij},$$

where $\hat{\pi}_i > \hat{\pi}_j$ means that node $i$ is successive to $j$ in the order. If $\hat{\pi}$ is the right topological order for $A$, then $D_{\mathrm{top}}(\hat{\pi}, A) = 0$. Else, $D_{\mathrm{top}}(\hat{\pi}, A)$ counts the number of edges that cannot be recovered due to the choice of topological order. For example, given a graph $X_1 \to X_2 \to X_3$ with adjacency $A$, the causal order $\hat{\pi} = \{1, 3, 2\}$ does not allow an edge $X_2 \to X_3$, and $D_{\mathrm{top}}(\hat{\pi}, A) = 1$. Given that Theorem 1 concerns the identifiability of the causal order, and our goal is to empirically support our theoretical findings, the topological order divergence is the right metric to monitor. In Figure 6 and Figure 7 we report the average $D_{\mathrm{top}}$ over 20 seeds, and the error bars are 95% confidence intervals.

**Random baseline.** The performance of our algorithm is compared with that of a random baseline: in particular, in the graph we report the mean accuracy of an algorithm that randomly sample a causal order among all possible permutations of the set $\{1, ..., d\}$, $d$ being the number of nodes. If the upper boundary of the 95% confidence intervals around the mean accuracy of our method are lower than the mean of the random baseline, that's statistically significant empirical evidence in support of our theory.

Next, we proceed to analyse the experiments. We separately consider the case of inference on linear and nonlinear structural causal models.

### F.5.1 EXPERIMENTS ON LINEAR SCMs

When synthetic data are generated according to a linear model $\mathbf{X} = A\mathbf{S}$ ($A$ being the mixing matrix), the Hessian of the log-likelihood is equal to the inverse of the covariance matrix $\Sigma_{\mathbf{X}}$ (the Hessian, in this case, takes the name of *precision matrix*). For this reason, in the linear setting, we replace the Stein gradient estimator of the Hessian with a simple approximation of the covariance $\Sigma_{\mathbf{X}}$ via averaging. The motivation is two-fold: *(i)* Hessian estimation via the Stein gradient is unstable as the dimension of the graph grows (see, e.g., (Montagna et al., 2023a)); *(ii)* the average estimator is much faster, which allows us to scale our experiments to higher dimensions. In the linear case, our method is similar to the BACKSHIFT algorithm (Rothenhäusler et al., 2015).

**Synthetic data generation.** We analyse the performance of Algorithm 1 on graphs with $\{10, 20, 50\}$ nodes, respectively with number of edges $\{10, 40, 100\}$. Graphs are generated via the Erdös–Rényi model (Erdos & Renyi, 1960). For each graph, we run experiments with $\{3, 6, 9\}$ environments.

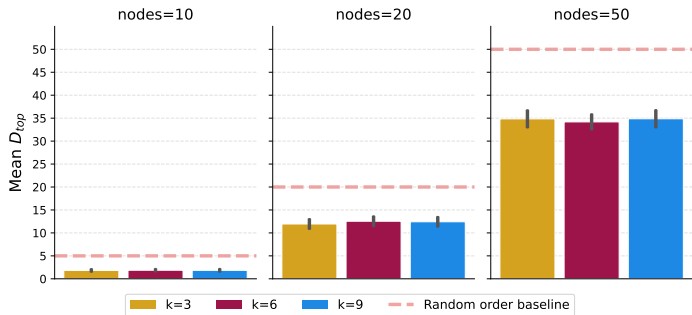

Figure 6: Mean $D_{\text{top}}$ (the lower, the better) of Algorithm 1 on data generated with a synthetic linear SCM and graphs with different number of nodes $(10, 20, 50)$. Error bars are $95\%$ confidence intervals. $k$ refers to the number of environments. We note that, in line with our theory, 3 environments are sufficient to infer causality much better than random.

Rescaling coefficients for the source variance are uniformly sampled between 2 and $\min(2|\mathcal{G}|, 10)$, $|G|$ being the number of nodes in the considered graph. A dataset from a single environment consists of 2000 i.i.d. samples. The linear regression coefficients are uniformly sampled from $[2, 5]$, and the sign of the coefficient is randomly flipped.

**Analysis of the experiments.**   In Figure 6 we see that even in high dimensions, our method can infer causality on linear Gaussian models with as few as three environments. In particular, on 10 nodes, the mean error is reduced by $\approx 75\%$ compared to the random baseline; on 20 nodes, we see improvements of $\approx 45\%$; on 50 nodes, the error decreases by $\approx 30\%$. It's remarkable how the method's accuracy does not improve with more than 3 environments. This is in line with our theory, which demonstrates that 3 sufficiently different environments guarantee identifiability of the causal graph.

### F.5.2   EXPERIMENTS ON NONLINEAR SCMs

We now consider the empirical performance of Algorithm 1 on nonlinear structural causal models with 5 nodes. With already 10 nodes, we observe that our method infers a causal order that is, on average, no better than random, suggesting that further research for a good algorithmic implementation of our theoretical findings is necessary. To put this in perspective, we remark the goal of our experiments, and more generally, of the paper: the contribution of our work is devoted to establishing novel identifiability results for causal discovery with multiple environments, leveraging the duality between ICA and structural causal models; on the contrary, the goal is not to present novel algorithmic solutions based on these results. With this in mind, we design Algorithm 1 as a simple implementation of the steps in the proof of Theorem 1; we do not claim that this is a good strategy beyond our purpose of validating the theory with toy examples. In fact, according to the literature and our experience, multi-environment causal discovery with ICA is a challenging problem on its own (see the discussion in Section B.2): as such, we leave it to future research. Our experiments only serve the purpose of demonstrating that our theoretical results and our proof techniques are correct. In line with this goal, we find that our method only requires 3 environments to infer causal directions significantly better than random on 5 nodes, even in challenging nonlinear scenarios.

**Synthetic data generation.**   We consider synthetic data generated with nonlinear structural causal models that are not identifiable from pure observations, and satisfy the assumptions of Theorem 1. In particular, given a variable $x_j$ and its parents $x_{\text{PA}_j}$, our mechanisms are defined as follows: first we define a *cause* random variable $c := \frac{1}{|\text{PA}_j|}\sum_{k \in \text{PA}_j} x_k$ as the mean of the parents; then, given the noise $s_j$, we consider the following causal mechanisms: *(i)* $x_j := \cos(c)s_j + \arctan(s_j)$; *(ii)* $\tanh(c)\arctan(s_j) + s_j^3$; *(iii)* $\sin(c) + \arctan(c)s_j + \cos(c)s_j^3$. Note that, differently from the experiments in Section 4 on bivariate graphs, we wrap the *cause* in trigonometric functions and avoid

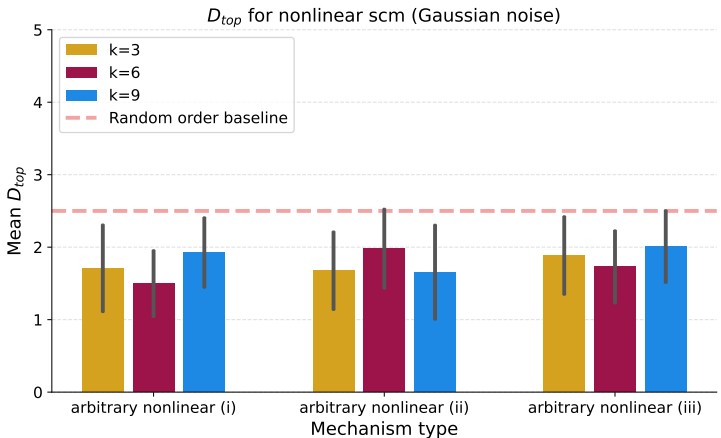

Figure 7: Mean $D_{\text{top}}$ (the lower, the better) of Algorithm 1 on data generated with a synthetic nonlinear SCMs with 5 variables. Error bars are 95% confidence intervals. $k$ refers to the number of environments. We note that, in line with our theory, 3 environments are sufficient to infer causality better than random, and adding environments does not decrease the error.

polynomials. This is to prevent the variance from growing polynomially in the causal direction (a well-known phenomenon in simulated SCMs (Reisach et al., 2021)), which we observed to cause all values in the Hessian of the log-likelihood to collapse to zero. Graphs are generated via the Erdös–Rényi model (Erdos & Renyi, 1960). For each graph, we run experiments with $\{3, 6, 9\}$ environments. The rescaling coefficients per-environment of the source covariance are uniformly sampled between 2 and 10. A dataset from a single environment consists of 2000 i.i.d. samples.

**Analysis of the experiments.** Figure 7 shows that, for structural causal models with 5 nodes, 5 edges and nonlinear mechanisms, information about the causal order can be inferred by our method: in particular, compared to a random baseline, whose expected $D_{\text{top}}$ is 2.5, our method with 3 environments yields improvements between $\approx 30\%$ (on nonlinear mechanisms of type *(i)*) and $\approx 25\%$ (for mechanisms of type *(iii)*). Notably, in line with our theory, adding environments does not decrease the average error across seeds, showing that only 3 sufficiently different environments are needed for inference.

## G   Assumptions deepdive

We present further discussion on the assumptions of our theory and potential extensions beyond them.

### G.1   Beyond Gaussianity

One of the key restrictions of our theory is that it requires the independent noise terms to be Gaussian. In the main paper, we discuss how this can be relaxed to noise distributions whose gradient of the log-likelihood has a critical point. Here, we expand on the discussion of Section 3.1 to illustrate the fundamental limit of our proof technique to address the case of general noise distributions. To begin, we provide a step-by-step mathematical intuition of why Gaussianity is crucial for our proof. The key ingredient of our theory is the analysis of the Hessian of the log-likelihood. By the chain rule of differentiation, it can be verified that the score function at a data point $\mathbf{x}$, under environment $i$, satisfies:

$$\nabla \log p^e(\mathbf{x}) = J_{\mathbf{f}^{-1}}(\mathbf{x})^T \nabla \log p^e(\mathbf{s}). \tag{34}$$

Applying once again the chain rule, one can easily verify the following expression of the Hessian of the log-likelihood:

$$J_{\mathbf{f}^{-1}}(\mathbf{x})^T D_{\mathbf{s}}^2 \log p^e(\mathbf{s}) J_{\mathbf{f}^{-1}}(\mathbf{x}) + D_{\mathbf{x}}^2 \log |J_{\mathbf{f}^{-1}}(\mathbf{x})| + \sum_{j=1}^{d} \partial s_j \log p^e(s_j) D^2 \mathbf{f}_j^{-1}(\mathbf{x}). \tag{35}$$

The information about the causal graph is contained in the product of Jacobians $J_{\mathbf{f}^{-1}}(\mathbf{x})^T D_{\mathbf{s}}^2 \log p^e(\mathbf{s}) J_{\mathbf{f}^{-1}}(\mathbf{x})$ (the diagonal Hessian in between doesn't play a significant role). To access this information from the Hessian of the log-likelihood, we need to get rid of:

1. The log-det term $D_{\mathbf{x}}^2 \log |J_{\mathbf{f}^{-1}}(\mathbf{x})|$;

2. The summation $\sum_{j=1}^d \partial s_j \log p^e(s_j) D^2 \mathbf{f}_j^{-1}(\mathbf{x})$.

Being the mechanisms $\mathbf{f}$ invariant across the environments, it is immediate to see that $\log |J_{\mathbf{f}^{-1}}(\mathbf{x})|$ vanishes in the difference $D_{\mathbf{x}}^2 \log p(\mathbf{x}) - D_{\mathbf{x}}^2 \log p^e(\mathbf{x})$. The assumption of Gaussianity, instead, is crucial to get vanishing summation: in fact, we know that the mean of the sources $\mathbf{s} = \mu_{\mathbf{S}}$ is a critical point of $\log p_{\mathbf{S}}$. This clarifies why the assumption of Gaussianity is crucial for our theory.

A natural question is whether our theory can extend to structural causal models with more general classes of noise distributions. Beyond density functions with a critical point, the answer is generally negative. To show why this is the case, we consider the exponential family, which encompasses a large class of common distributions. Let $\mathbf{S}$ distributed according to the exponential family with the vector of parameters $\theta$ (in the Gaussian case, $\theta = (\mu_{\mathbf{S}}, \Sigma_{\mathbf{S}})$). Then:

$$\log p(\mathbf{s}) = \log h(\mathbf{s}) + \eta(\theta) \cdot T(\mathbf{s}) - A(\eta), \tag{36}$$

where $h(\mathbf{s})$ is the so called *base measure*, $\eta(\theta)$ is the vector of the *natural parameters*, $T(\mathbf{s})$ is the vector of *sufficient statistics*, and $A(\eta)$ is the *partition function*. Now, assume that, akin to the Gaussian case, we define auxiliary environments (cf. Equation (4)) by changing $\theta^e$ parameters for each environment $e$. The difference of the score of the observed variables $\mathbf{x}$, in this case, becomes:

$$\nabla \log p(\mathbf{s}) - \nabla \log p^e(\mathbf{s}) = T(\mathbf{s}) \cdot (\eta(\theta) - \eta(\theta^e)).$$

Assuming that $\theta \neq \theta^e$ in each component, we get that the score of the sources vanishes if and only if $T(\mathbf{s}) = 0$ or orthogonal to $\eta(\theta) - \eta(\theta^e)$. Clearly, orthogonality can not be enforced unless we carefully craft the intervention on $\theta$. It remains to consider whether the $T(\mathbf{s})$ vanishes at any point. A simple inspection of the sufficient statistics of the density functions in the exponential family reveals that this is often not the case.

The takeaway of our discussion are: *(i)* that, as far as it concerns our methodology, vanishing gradient of the log-likelihood at one point at least is *necessary*; when this is not the case, we can not extract the product of Jacobian matrices (hence, the DAG information) from the Hessian of the log-likelihood. This is in line with previous work (Montagna et al., 2023a; 2025), showing that the Hessian matrix can only inform about the equivalence class of the ground truth graph. *(ii)* For wider classes of noise distributions, in general, we can not hope that the vanishing gradient condition is satisfied. Thus, extension of our results requires substantial additional research in terms of proof techniques.

## G.2 BEYOND CAUSAL SUFFICIENCY

In this section, we address the question of whether our methodology can be adapted to demonstrate the identifiability of parts of the causal graph in potentially confounded scenarios. The duality between ICA and causal discovery that is key to this paper remains relevant even in this scenario. This was explicitly highlighted in Ding et al. (2019), where, in the context of linear SCMs with latent confounders, causal discovery is phrased and analysed as an overcomplete ICA problem. For general nonlinear structural causal models, the presence of latent confounders induces an ICA model $\mathbf{X} = \mathbf{f}(\mathbf{S})$ with $\mathbf{f} : \mathbb{R}^{d_s} \to \mathbb{R}^{d_x}$ and $d_s > d_x$. First, we discuss why our proof technique can not be generalized to this scenario when $\mathbf{f}$ is nonlinear. Then, we show that in the case of linear structural causal models, our findings can be used to derive known theory of identifiability of SCMs without causal sufficiency.

We remind that the key theoretical result that enables identifiability in our setting (Theorem 1) is Lemma 1, which we report below.

**Lemma 1.** *Let $\mathbf{x} = \mathbf{f}(\mathbf{s}) = \widehat{\mathbf{f}}(\hat{\mathbf{s}})$, where $\mathbf{s} = \mu_{\mathbf{S}}$. Let Assumptions 1,2 and 4 satisfied. Then:*

$$\sum_{e \in \mathcal{E}_1} D_{\mathbf{x}}^2 \log p(\mathbf{x}) - D_{\mathbf{x}}^2 \log p^e(\mathbf{x}) = J_{\mathbf{f}^{-1}}(\mathbf{x})^T \Omega_1 J_{\mathbf{f}^{-1}}(\mathbf{x}) = J_{\widehat{\mathbf{f}}^{-1}}(\mathbf{x})^T \widehat{\Omega}_1 J_{\widehat{\mathbf{f}}^{-1}}(\mathbf{x}) \tag{10}$$

$$\sum_{e \in \mathcal{E}_2} D_{\mathbf{x}}^2 \log p(\mathbf{x}) - D_{\mathbf{x}}^2 \log p^e(\mathbf{x}) = J_{\mathbf{f}^{-1}}(\mathbf{x})^T \Omega_2 J_{\mathbf{f}^{-1}}(\mathbf{x}) = J_{\widehat{\mathbf{f}}^{-1}}(\mathbf{x})^T \widehat{\Omega}_2 J_{\widehat{\mathbf{f}}^{-1}}(\mathbf{x}) \tag{11}$$

Clearly, the result above relies on the invertibility of the causal mechanism $\mathbf{f}$. Moreover, it is easy to show that $\Omega_i, \widehat{\Omega}_i$ are diagonal, which is key to the proof of Theorem 1. Unfortunately, in overcomplete ICA:

1. It is trivial that $\mathbf{f}$ is not invertible.

2. Less trivially, computations based on the coarea formula (Negro, 2021) show that $\Omega_i, \widehat{\Omega}_i$ are non-diagonal.

From this, we conclude that generalizing our method for arbitrary nonlinear and confounded SCMs is not a feasible route, and more elaborate tools and ideas are required. We note that, exceptionally, the Hessian of the log-likelihood is still informative about the causal graph in case of linear and overcomplete SCMs: in fact, its inverse is the covariance of the data, namely, $(D_\mathbf{x}^2 \log p(\mathbf{x}))^{-1} = \Sigma_\mathbf{X} = A\Sigma_\mathbf{S} A^T$, for a structural model of the form $\mathbf{X} = A\mathbf{S}$, with $A$ rectangular, wide, matrix. Notably, in this setting, rank constraints and trek separations (Sullivant et al., 2010) are informative about the causal graph.

## H    ADDITIONAL CONTENT

In this section, we collect some useful results and notes relevant to the main paper.

### H.1    GRAPH THEORY

**Directed graphs and DAGs.**    Let $X_1, \ldots, X_d$ be a vector of random variables. A graph $\mathcal{G} = (\{X_i\}_i^d, E)$ consists of a vertex set $\{X_i\}_i^d$ and an edge set $E$. We recall a few basic notions for directed graphs.

A *directed edge* $X_i \to X_j$ indicates that $X_i$ is a *parent* of $X_j$ (and $X_j$ a *child* of $X_i$). $\mathrm{PA}_i \subset [d]$ denotes the index of the parent nodes of $X_i$ in the graph $\mathcal{G}$, $\mathrm{CH}_i \subset [d]$ denotes the children. A *path* in $\mathcal{G}$ is a sequence of at least two distinct vertices $\pi = X_{i_1}, \ldots, X_{i_m}$ such that each consecutive pair $X_{i_k}$ and $X_{i_{k+1}}$ is joined by an edge for $k = 1, \ldots, m-1$. If every edge along the path is oriented forward, $X_{i_k} \to X_{i_{k+1}}$, we call it a *directed path*; then $X_{i_1}$ is an *ancestor* of $X_{i_m}$ and $X_{i_m}$ a *descendant* of $X_{i_1}$.

### H.2    FROM SCM TO ICA MODELS

Equation (2) claims that structural causal models can be expressed in the form of ICA models. Here, we show how this can be achieved. Consider a set of causal variables $\mathbf{X} = (X_i)_{i=1}^d$, and without loss of generality, assume that the causal order is $1, \ldots, d$. According to Equation (1), for each $i = 1, \ldots, d$, we have:

$$X_i := F_i(\mathbf{X}_{\mathrm{PA}_i}, S_i),$$

with $\mathbf{S} = (S_i)_{i=1}^d$ the vector of mutually independent noise terms. An inductive argument shows the existence of a function $f_i : \mathbf{S}_{\mathrm{AN}_i} \mapsto X_i$, where $\mathrm{AN}_i$ denotes the indices of the ancestor nodes of $X_i$ in the causal graph. Given the causal order $1, \ldots, d$, the base case is given for $X_1 := F_1(S_1)$, such that $f_1 := F_1$. The inductive step is as follows: assume that there is $n < d$ such that $X_i = f_i(\mathbf{S}_{\mathrm{AN}_i}, S_i)$ for all $i = 1, \ldots, n$. Then, there is a map $\mathbf{S}_{[n]} \mapsto \mathbf{X}_{[n]}$. The causal order $1, \ldots, d$ implies $\mathrm{AN}_{n+1} \subset [n]$, so that there is a map $\mathbf{S}_{[n]} \mapsto \mathbf{X}_{\mathrm{AN}_{n+1}}$: given that $\mathrm{PA}_{n+1} \subseteq \mathrm{AN}_{n+1}$, there is a map $g : \mathbf{S}_{[n]} \mapsto \mathbf{X}_{\mathrm{PA}_{n+1}}$: from the structural equation $X_{n+1} := F_{n+1}(\mathbf{X}_{\mathrm{PA}_{n+1}}, S_{n+1}) = F_{n+1}(g(\mathbf{S}_{\mathrm{AN}_{n+1}}), S_{n+1})$, we conclude that there is $f_{n+1} : \mathbf{S}_{\mathrm{AN}_{n+1}}, S_{n+1} \mapsto X_{n+1}$. Then, we define $\mathbf{f} := (f_i)_{i=1}^d$ and find

$$\mathbf{X} = \mathbf{f}(\mathbf{S}).$$

An important note is that the DAG structure of the causal graph is reflected in the Jacobian of the mixing function $\mathbf{f}$, which can be shown to be lower triangular.

### H.3    HESSIAN OF THE LOG-DENSITY OF INDEPENDENT RANDOM VARIABLES

In the main paper we mention that the $\Omega_1, \Omega_2$ matrices defined in Equation (9) are diagonal; here, we discuss why this is true. More generally, it is well known that for a vector of independent random

variables $\mathbf{Z} \in \mathbb{R}^d$ with density $p$, the following holds:

$$\frac{\partial^2}{\partial Z_i \partial Z_j} \log p(\mathbf{Z}) = 0 \iff Z_i \perp\!\!\!\perp Z_j | \mathbf{Z} \setminus \{Z_i, Z_j\}, \tag{37}$$

where $Z_i \perp\!\!\!\perp Z_j | \mathbf{Z} \setminus \{Z_i, Z_j\}$ indicates that $Z_i, Z_j$ are independent conditional on all the remaining random variables in the vector $\mathbf{Z}$. This result was shown in Lin (1997) and Spantini et al. (2018) (Lemma 4.1) and extensively adopted in the context of causal discovery (e.g., Montagna et al. (2023a; 2025)). By Equation (37) it is immediate to see that independence of $\mathbf{Z}$ entries implies that $D_{\mathbf{Z}}^2 \log p(\mathbf{Z})$ is diagonal.

### H.4 MEASURE THEORETIC ARGUMENTS IN SUPPORT OF THE ASSUMPTIONS

First, we show that Assumption 5 generically holds.

**Proposition 3** (Assumption 5 holds almost surely). *Let $L_e = \mathrm{diag}(\lambda_1^e, \ldots, \lambda_d^e)$ and $\lambda_j^e \neq 0$, $e = 1, \ldots, k$. Assume the joint law of the array $\Lambda = (\lambda_j^e)_{j \in [d], e \in [k]}$ is absolutely continuous with respect to Lebesgue measure on $\left(\mathbb{R} \setminus \{0\}\right)^{dk}$. Then, with probability one over the draw of $\Lambda$: for every $j \in [d]$,*

$$\sum_{e \in [k]} \frac{1}{(\lambda_j^e)^2} \neq k.$$

*Proof.* Fix $j \in [d]$. Write $k = |\mathcal{E}_l|$ and $\lambda := (\lambda_j^e)_{e \in [k]} \in (\mathbb{R} \setminus \{0\})^k$. Consider the smooth map $F : (\mathbb{R} \setminus \{0\})^k \to \mathbb{R}$,

$$F(\lambda) = \sum_{r=1}^{k} \lambda_r^{-2} - k.$$

Its gradient is $\nabla F(\lambda) = (-2\lambda_1^{-3}, \ldots, -2\lambda_k^{-3}) \neq 0$ on the domain, so $0$ is a regular value. By the regular level–set theorem, $F^{-1}(0)$ is a $(k-1)$-dimensional embedded submanifold of $\mathbb{R}^k$ and hence has Lebesgue measure zero. Because the $k$-tuple $\lambda = (\lambda_j^e)_{e \in [k]}$ has a distribution absolutely continuous with respect to Lebesgue measure, we get

$$\mathbb{P}\left(\sum_{e \in \mathcal{E}_l} \frac{1}{(\lambda_j^e)^2} = k\right) = 0.$$

Taking the finite union over $j = 1, \ldots, d$ preserves measure zero, so with probability one none of these equalities occurs. $\qquad\square$

Next, we show that the assumption of pairwise distinct $\{(\Omega_1 \Omega_2^{-1})_i i\}_{i \in [d]}$ elements (definition at Equation (9)) generically holds.

**Proposition 4** (Pairwise distinct diagonal ratios hold almost surely). *Let $\mathcal{E}_1, \mathcal{E}_2 \subset [k]$ with $k \geq 3$. For each environment $e$ let $L_e = \mathrm{diag}(\lambda_1^e, \ldots, \lambda_d^e)$ with $\lambda_j^e \neq 0$. Assume the joint law of the array $\Lambda = (\lambda_j^e)_{j \in [d], e \in [k]}$ is absolutely continuous with respect to Lebesgue measure on $\left(\mathbb{R} \setminus \{0\}\right)^{dk}$. Suppose moreover that $\Omega_\ell$ is diagonal with entries*

$$(\Omega_\ell)_{jj} = \frac{1}{\sigma_j^2}\left(\sum_{e \in \mathcal{E}_\ell} (\lambda_j^e)^{-2} - |\mathcal{E}_\ell|\right) \neq 0. \qquad \ell \in \{1, 2\}, \ j \in [d],$$

*Then, with probability one over the draw of $\Lambda$, $\Omega_1$ is invertible and the diagonal entries of $\Omega_1^{-1}\Omega_2$ are pairwise distinct.*

*Proof.* Write

$$(\Omega_1^{-1}\Omega_2)_{jj} = \frac{\sum_{e \in \mathcal{E}_2} (\lambda_j^e)^{-2} - |\mathcal{E}_2|}{\sum_{e \in \mathcal{E}_1} (\lambda_j^e)^{-2} - |\mathcal{E}_1|} =: \frac{B_j}{A_j}, \qquad A_j := \sum_{e \in \mathcal{E}_1} (\lambda_j^e)^{-2} - |\mathcal{E}_1|, \ B_j := \sum_{e \in \mathcal{E}_2} (\lambda_j^e)^{-2} - |\mathcal{E}_2|.$$

By Proposition 3, $A_j \neq 0$ and $B_j \neq 0$ for all $j$ with probability one, such that $\Omega_1$ is invertible.

Fix $j \neq \ell$. The collision event $(\Omega_1^{-1}\Omega_2)_{jj} = (\Omega_1^{-1}\Omega_2)_{\ell\ell}$ is equivalent to

$$\frac{B_j}{A_j} = \frac{B_\ell}{A_\ell} \quad \Longleftrightarrow \quad F_{j\ell}(\Lambda) := A_j B_\ell - A_\ell B_j = 0.$$

Let $t_h^e := (\lambda_h^e)^{-2}$ and view $F_{j\ell}$ as a smooth function of the $2k$ variables $\{t_j^e\}_{e \in [k]} \cup \{t_\ell^e\}_{e \in [k]}$. For any fixed $e_0 \in \mathcal{E}_1$,

$$\frac{\partial F_{j\ell}}{\partial t_j^{e_0}} = \frac{\partial A_j}{\partial t_j^{e_0}} B_\ell - A_\ell \frac{\partial B_j}{\partial t_j^{e_0}} = 1 \cdot B_\ell - A_\ell \cdot 0 = B_\ell.$$

Since $B_\ell \neq 0$, we have $\nabla F_{j\ell} \neq 0$ on the set under consideration, so 0 is a regular value of $F_{j\ell}$. By the regular level-set theorem, the set $\{F_{j\ell} = 0\}$ is a $(2j - 1)$-dimensional embedded submanifold of $\mathbb{R}^{2k}$, hence it has Lebesgue measure zero. Because the law of $\Lambda$ is absolutely continuous w.r.t. the Lebesgue measure,

$$\mathbb{P}\left((\Omega_1^{-1}\Omega_2)_{jj} = (\Omega_1^{-1}\Omega_2)_{\ell\ell}\right) = 0.$$

Taking the finite union over all pairs $j \neq \ell$ yields that, with probability one, no two diagonal entries coincide; that is, $\{(\Omega_1^{-1}\Omega_2)_{jj}\}_{j=1}^d$ are pairwise distinct. $\qquad\square$

## H.5    FIXED MECHANISMS ENVIRONMENTS IN REAL-WORLD DATA

In this section we briefly discuss the assumption of *fixed mechanisms* across environments that is formalized in the *invariance principle* (Section 2.2): given two environments $\mathbf{X}^e = \mathbf{f}(\mathbf{S}^e)$, $\mathbf{X}^{e'} = \mathbf{f}(\mathbf{S}^{e'})$, they share the same causal mechanism $\mathbf{f}$. In particular, we present examples from the domain of single-cell and gene perturbation causality studies where multiple environments with fixed mechanisms are commonly hypothesized. This suggests that our modeling assumptions, hence our theory, have practical relevance.

Liu et al. (2025) and (Lopez et al., 2023) assume an SCM and explicitly model gene and single-cell (respectively) perturbations as changes in the distribution of causal variables, while leaving all SCM mechanisms fixed. Similarly, but without an explicit assumption of a structural causal model, Zhang et al. (2023) consider interventions on latent factors that leave causal mechanisms unchanged. Meinshausen et al. (2016) studies the problem of gene perturbation through the Invariance Causal Prediction framework (Peters et al., 2015): in this context, they discuss the example of environments defined with fixed causal mechanisms and noise variance affected by a multiplier that is environment dependent. This is precisely in line with the modelling assumptions of our theory.

