# OpenReview forum: "On the Identifiability of Causal Graphs with the Invariance Principle"
_ICLR.cc/2026/Conference — ICLR 2026 Poster_

### Official Review · Reviewer_5jiF · 2025-10-16

**Soundness:** 3
**Presentation:** 3
**Contribution:** 2
**Rating:** 4
**Confidence:** 3

**Summary:**

This paper studies the condition for the unique acyclic causal graph identification from multiple environments.  Under assumptions around nonlinear ICA, the authors show that only two sufficiently different environments is enough for the identifiability.  The technique is by probing the Jacobian mixing functions' support.

**Strengths:**

1. The **main theorem is well motivated and looks correct to me;** I didn't check for details though.

2. The authors **do a good job in discussing the connection between causal discovery and independent component analysis (ICA),** and explain why causal discovery can be easier than full ICA recovery -- one cares the support of the mixing matrix at one point, while another cares the independent sources recovery at each point.

3. **The overall presentation is good.**  The assumptions are clearly listed, and the writing is still clear with heavy notations.

**Weaknesses:**

1. **My major concern is that the assumptions are way too strong, untestable, and goes a bit against the core goal interventional causal discovery:**
- Though the conclusion "only two environments needed; this number does not depend to the number of vertices in graph" may sound appealing at first, it comes with the price that the causal mechanisms (the entire mixing function) must remain fixed across all environments, the noise components are required to be Gaussian, and only variance rescaling is allowed between interventions.
- These conditions are too strong merely for the goal of "two environments needed".  **This goal itself somewhat goes against the goals for both interventional causal discovery and the characterization of environments needed:**
   - **For interventional causal discovery,** the main idea is to fully use the available information from multiple environments' data (while allow the changes across environments to be arbitrarily flexible).  It is good as long as we can identify more than from only one environment (e.g.,  the CPDAG).  The goal is not the exact DAG recovery at any cost, but rather to make full use of available information in practice. Otherwise, if the goal is just exact DAG recovery even with model misspecification, why bother to use the difficult estimations involved in this work, instead of just running LiNGAM on one domain?
   - **For characterization of environments needed** for exact DAG recovery, or other characterizations to the sizes of the equivalence class under interventions: the main purpose is to characterize the internal randomness from data induced by the causal graph, and to help design experiments -- e.g., about how many interventions and which targets are needed for the exact DAG recovery, so as to do the exact effect estimation.  The dependence of this number to the graph size is not a bad thing.  And again, experiments in real world need to be flexible and the strict assumptions in this work are not desired.

2. **Assumptions can be stated more clearly:**

For instance, in Assumption 2 about noise rescaling:
 - Is mean shift also allowed, as long as the variance changes?
 - Does L have to be diagonal, or is it enough to be orthogonal against Si's diagonal variances?

3. **Experiments are limited to bivariate case with synthetic data.**  This is minor and acceptable though, given the theoretical focus of this paper.  The authors also acknowledge this in the appendix, which is fair.

**Questions:**

Could the authors elaborate more on the role of acyclicity in their framework?

Given the close connection between causal discovery and ICA, I am curious whether acyclicity is truly essential for the results in this work.  For instance, In the linear non-Gaussian case, as shown in https://arxiv.org/abs/1206.3273, the presence of cycles does not create technical difficulties.  One can simply run ICA and interpret the resulting demixing matrix differently (not necessarily to be lower triangular;  as long as it's nonzero entries on diagonal).

Is the same reasoning applicable in the nonlinear ICA setting here, or does acyclicity play some more essential roles here?


And minor, at L133 notations, "Also, we use [d] := {1,...,d}." should be stated before being used at "supp(M) := {(i,j)|i∈[m],j ∈[n] ".

---

> ### Author Response · Authors · 2025-11-22
>
> We thank the reviewer for their time and feedback on our work.
>
> Two major concerns are raised by the reviewer, which we address below.
>
> - There could be a misunderstanding about the core contribution of our work: the **main concern of the reviewer is that our paper goes against the goal of interventional causal discovery.** To argue this, the reviewer writes that the goal of interventional CD is, e.g., to
>
>     > [characterize] how many interventions and which targets are needed for the exact DAG recovery.
>     >
>
>     We compare this to the content of our paper, that is as follows: taking causal discovery as a special case of ICA, we characterize how many interventions (in the form of environments) and which targets (in terms of noise terms) are needed for the exact DAG recovery: **our contribution exactly matches the goal of interventional causal discovery according to the reviewer**.
>
> - The other main concern raised by the reviewer is that assumptions are too strong.
>     - **Fixed mechanism assumption.**  We argue that the strength of assumptions is problem-dependent: the assumption of **fixed mechanisms across environments is common in single-cell/gene perturbation causality studies**, meaning that our setting is relevant to real-world problems. Liu et al. (2025), Lopez et al. (2023), and Dong et al. (2025) model gene and single-cell perturbations as changes in random variables distributions, while leaving all mechanisms fixed.  Meinshausen et al. (2016) discuss environments defined by rescaling noise variance while keeping fixed causal mechanisms: this is **exactly our setting**. We added a discussion and relevant bibliography on this in Appendix G.5 of the paper.
>     - **Gaussianity.** In reference to the concerns about the assumption of Gaussian noise, we note the following:
>         - In the paper, we show that this is sufficient but not necessary, as well as ways to relax it.
>         - **Other works that have more permissive noise definitions**, notably the Invariant Causal Principle paper from Peters et al. (2015),  **have other strong and untestable requirements like linearity of the SCM**. With this, we highlight that theoretical results come with a trade-off with assumptions: our paper provides a novel characterization of this trade-off, often weakening assumptions (e.g., compared to Peters et al. (2015) and Heinze et al. (2018) in terms of mechanism restrictions, Monti et al. (2019) in terms of the number of necessary environments).
>
> ---
>
> **Clarity of assumption 2.**
>
> - **Does L have to be diagonal?** Yes. This is precisely written: $L_i = \operatorname{diag}(\lambda_1^i, …, \lambda_d^i)$, $\lambda^i_j \neq 0$.
> - **Are mean shifts allowed?** No. We write that new environments are given by  $\mathbf S^i = L_i \mathbf S^0$, hence we exclude shifts.
>
> ---
>
> **Experiments on multivariate graphs**. Experiments on multivariate graphs are added in **Appendix E.5** of the paper (please refer to the general response of this rebuttal for details).
>
> ---
>
> **Question on acyclicity**. Thanks for the good reference. In our work, acyclicity only serves the purpose of removing the permutation indeterminacy in a LiNGAM-like fashion. But, as pointed out by the reviewer, getting rid of the assumptions would not present technical difficulties, much like in https://arxiv.org/abs/1206.3273.
>
> ---
>
> We thank the reviewer for their feedback. In the new version of the manuscript, we have added the requested experiments (Appendix E.5). We responded to all the concerns and questions of the review. **In light of the new results and clarifications, we kindly ask the reviewer to consider increasing their rating.** We remain available for further discussion.

---

> ### Comment · Reviewer_5jiF · 2025-11-27
>
> Thank the authors for the response and the added experiments on multivariate cases.
>
> Regarding my major concern of "this work going a bit against the core goal of interventional causal discovery", I believe I do not have a misunderstanding.
>   - I acknowledge that the forms are the same, e.g., "characterizing how many interventions and which targets are needed for identifying something". I also acknowledge the authors in connecting causal discovery with ICA.
>   - But what I truly doubt is whether there is any practical applicability for this work, in giving an unnecessarily strong result of "exact DAG recovery from only two environments", but at the cost of very restrictive, uninterpretable, and arguably unnatural assumptions, such as "the entire mixing function remaining fixed" and "the only change being the noise variance and this change being sufficient".
>   - For interventional causal discovery, the most common scenario is that there are multiple environments (more than just 2), each environment undergoes some local component change (instead of just a global noise rescaling), and one seeks identifying an equivalence class that uses all available information (doesn't have to be an exact DAG).
>   - In contrast, if one only seeks to discover an exact DAG, why do they bother to use this work, instead of the many alternatives that even do not require interventions, such as LiNGAM and ANM?  In either case, I doubt that there is model misspecification so..
>   - Overall, if this paper is targeted for the purpose of "showing some identifiability result of nonlinear ICA with only two environments", I may have raised my score to 6;  not higher because 1) the assumptions are still strong for nonlinear ICA, and 2) the proof techniques are those commonly used ones.
>   - However, since the paper is now targeted for "identifiability with causal discovery", I can only remain my score of 4, because for this purpose, more characterization of the identifiability (at the equivalence class level) under less restricted interventions are expected. The current formulation seems doubtful to have any practical applicability.

---

> > ### Author Response · Authors · 2025-11-30
> >
> > We thank the reviewer for engaging in the discussion. We believe our rebuttal largely addressed all three weaknesses highlighted in the initial review:
> >
> > - Concerns about the clarity of Assumption 2 are addressed.
> > - Concerns about the absence of multivariate experiments is addressed by the new empirical results (Appendix E.5)
> > - In their last response, the reviewer acknowledges that the content of our paper is in line with what, in their words, is one of the goals of interventional causal discovery; the initial perceived misalignment was one of their main concerns.
> >
> > **In line with this summary, we believe that our rebuttal largely clarified all points raised by the reviewer.** However, in their latest response, they provide some motivations for not raising the score, one of which is the **claim that our proof technique is not new.**  In this regard:
> >
> > 1. We are surprised by this new concern, as it was not part of the original review.
> > 2. Moreover, **we disagree with this claim.** In case we mistake, we kindly ask the reviewer to provide a reference where the difference in the Hessian of the log-likelihood is used to infer the product of Jacobians $J_{\mathbf f^{-1}}^T J_{\mathbf f^{-1}}$, and this is, in turn, used to characterise the identifiability of the causal graph. To the best of our knowledge, this doesn’t appear in any previous work.
> >
> > ---
> >
> > We remain available for further discussion to clarify any remaining doubts.

---

### Official Review · Reviewer_eLzQ · 2025-10-27

**Soundness:** 4
**Presentation:** 3
**Contribution:** 4
**Rating:** 10
**Confidence:** 4

**Summary:**

This paper proposes novel results for causal discovery from multi-environmental data, by relying on the relationship between representational identifiability and the identifiability of the causal graph. Their resulst provides novel insights into this relationship by characterizing the complexity of the causal discovery task in terms of the sufficient variability conditions prevalent in the ICA and CRL literatures.

Although the paper uses similar proof techniques as prior work (relying on the score and the support of the Jacobian), I believe their insight is extremely important (not to mention that it works for Gaussian sources, which is often a barraier to identifiability). The authors provide synethetic experiments to corroborate their findings (that causal discovery is independent of the number of performance, when their assumptions are met).

I have a few minor remarks for improvement (see below); nonetheless, **I am strongly in favor of the acceptance of the paper, and would recommend it for an oral if my concerns are addressed.**

**Strengths:**

- The paper is generally well-written
- The assumptions are clearly stated
- The theoretical results are well explained and contextualized (the "relation with ICA identifiability paragraph is very instructive")
- The experiments back up the theoretical findings (and even provide insights into how robust the findings are, i.e., when some of the assumptions are violated - I especially liked the ones with gamma distribution and vanishing/not vanishing gradients in E.4)
- The topic is important, especially that it shows a fundamental (and quantifiable) connection between representation identifiability (ICA) and causal discovery - in terms of the number of environments

**Weaknesses:**

My concerns are mainly about phrasing and writing; thus, minor (also see my clarifying questions below).

- abstract: please make it clear *two environments* is your best-case scenario. As of now, the current phrasing in the abstract might be misleading.
- L226: please put a reference where you discuss why Assm.4 is needed
	- In general, I'd consider putting all assumptions into a single listing
- Lem. 1.: for insights about the score function, there was previous work by Burak Varici and collaborators, I strongly recommend citing their papers.

**Questions:**

- What does "distribution of a structural causal model" mean in the abstract?
- L046: How do you prove that your results holds for arbitrary SCMs?
- L363: Could you please explain why the SCMs you used cannot be reparametrized as PNL or location-scale models?

---

> ### Author Response · Authors · 2025-11-22
>
> We thank the reviewer for the enthusiastic response! We are deeply grateful for the appreciation. In what follows, we address the points raised in the weaknesses section.
>
> - **Abstract.** In the revised paper, we explicitly added “*(in the best case)* only two environments”.
> - **Citing Varici.** Good point, we added the citation right after our Lemma 1.
> - **L226 reference.**  Done, thanks for the suggestion.
>
> ---
>
> **Questions.** We address the questions formulated in the review.
>
> - *What does "distribution of a structural causal model" mean in the abstract?* We mean the observational distribution entailed by the structural causal model. In the updated paper, we write “distribution *induced* by a structural causal model”.
> - *L046: How do you prove that your results hold for arbitrary SCMs?* In response to this question, we provide an intuition behind Lemma 1, the main result that enables Theorem 1.
>
>     > The main object of interest is the Hessian of $\log p^i(x)$ for each environment $i$. For a generic ICA model, iterated application of the chain rule of differentiation yields $D^2 \log p^i(\mathbf x)$ equals
>     >
>     >
>     > $J_{\mathbf f^{-1}}^T(\mathbf x)  D^2 \log p^i(\mathbf s) J_{\mathbf f^{-1}}(\mathbf x) + D^2 \log |J_{\mathbf f^{-1}}^T(\mathbf x)| + \sum_{j=1}^d \partial s_j \log p^i(s_j) D^2 \mathbf f^{-1}_j(\mathbf x)$
>     >
>     > The key idea is that the product $J_{\mathbf f^{-1}}^T(\mathbf x)  D^2 \log p^i(\mathbf s) J_{\mathbf f^{-1}}(\mathbf x)$ carries the information about the graph (the second-order derivative in between is a diagonal matrix, and doesn’t affect our analysis). Lemma 1 tells us how to access the quantity of interest $J_{\mathbf f^{-1}}^T J_{\mathbf f^{-1}}$:
>     >
>     > - The difference of Hessians between different environments cancels the log-det term: this is because $\mathbf f$ does not depend on the environments, hence is constant in the different Hessian expressions.
>     > - By considering $\mathbf s = \mu_{\mathbf S}$ (i.e., sources at the mean), Gaussianity guarantees that the summation term vanishes.
>     >
>     > Then, the difference of Hessians leaves us with the desired expression. This is a breakdown of Lemma 1, which contains the main intuitions behind the theory. The proof of Theorem 1 relies on technical (linear) algebraic manipulations of the result in the Lemma.
>     >
> - *L363: Could you please explain why the SCMs you used cannot be reparametrized as PNL or location-scale models?* To illustrate the concept, we show that  $x_1 := s_1$, $x_2 = s_1^2\arctan(s_2)+s_2^3$ used in our experiments can not be reparametrized to an LSNM.
>
>     > Consider the SCM $X_1 := S_1, X_2 := X_2(S_1, S_2) := S_1^2 \arctan(S_2) + S_2^3$ with independent noises $S_1 \perp S_2$. Any location–scale noise model with a single parent $X_1 \to X_2$ has the form
>     >
>     >
>     > $X_2 :=  X_2(\hat S_1, \hat S_2) := f(X_1) + g(X_1)\hat S_2$
>     >
>     > with $\hat S_1 \perp \hat S_2$. For such models, for any distinct $\hat S_1$ values $a,b$ and any distinct $\hat S_2$ values $u,v$:
>     >
>     > $R(u,v) = \frac{X_2(a,u) - X_2(a,v)}{X_2(b,u) - X_2(b,v)} = \frac{g(a)}{g(b)},$
>     >
>     > which is constant in $(u,v)$. Hence, a necessary condition for LSNM is $R(u,v)$ constant for each pair $a,b$. In our SCM, however, the same ratio becomes
>     >
>     > $R(u,v) = \frac{a^2(\arctan u - \arctan v) + (u^3 - v^3)}{b^2(\arctan u - \arctan v) + (u^3 - v^3)}.$
>     >
>     > For fixed $a \neq b$ this clearly depends on $(u,v)$; for instance with $a=1$, $b=2$ we obtain $R(1,0)\approx 0.43$ and $R(2,0)\approx 0.73$. Hence no representation of the form $f(X_1)+g(X_1)\hat S_2$ exists, and in particular this SCM cannot be reparametrized into a location–scale noise model.
>     >
>
>     Similar reasonings support our claim that the SCMs we used can not be reparametrized as PNL or location-scale models.

---

> > ### Comment · Reviewer_eLzQ · 2025-11-24
> >
> > Thank you for your detailed response, the updated submission is much better!

---

### Official Review · Reviewer_ZrxN · 2025-10-31

**Soundness:** 2
**Presentation:** 3
**Contribution:** 2
**Rating:** 2
**Confidence:** 3

**Summary:**

This paper considers the problem of recovering the causal graph in nonlinear structural causal model based on nonlinear independent analysis. Specifically, unlike previous results on causal discovery using nonlinear ICA, where the number of environments is required to be proportional to the number of sources for identifiability of the full model, the authors show that, if the task is to only recover the causal relations, under certain assumptions on the data generating mechanism, only three environments are required. Further, the authors provide an algorithm for graph identifiability based on estimation of the Jacobin matrix of the mixing function, and demonstrated the effectiveness of the proposed method through simulations based on synthetic data.

**Strengths:**

1. The problem formulation is clear. Specifically, the authors clearly state that the task is to identify the causal graph, which is different from causal discovery where the task is to identify the full causal model.
2. The theoretical results are clearly explained with detailed proofs.

**Weaknesses:**

1. Assumption 2 is very restrictive, making the theoretical results not novel. Specifically, Assumption 2 requires that the noises across the environments are "rescaled", i.e., $s_i^{k}=\lambda_i^{(k)} s_i^{0}$ for all $i,k$, where $s_i^k$ represents the noise term $s_i$ in environment $k$. This is a very strong assumption and asserts dependencies among the noises across environments. On the contrary, most of the existing results on nonlinear ICA only assumes that the noises are jointly gaussian with environemnt-specific covariance matrix.
2. The setting in the simulation results is over-simplified, which only considers the graph with two variables. In this case, the graph structure only has three possible choices, making the recovery task much easier. It would be better if the author could test the performance with more observed variables (say around 5). Also, it lacks comparison with baseline methods that use nonlinear ICA for causal discovery.

**Questions:**

1. How are $L_i$ selected in the numerical simulation? Specifically, in the settings with 6 and 9 environments, are $L_i$ different across all environments?
2. Since the task is to recover the graph structure instead of the causal model, can the theoretical results provided in this paper be extended to the case where the causal graphs across environments are the same but the functional mechanisms are different? Note that some of the the existing works allow for different mechanisms across environments, such as Jaber et al. (2020).

---

> ### Author Response · Authors · 2025-11-22
>
> We thank the reviewer for their feedback and time dedicated to our work. Below, we address the points raised in the review’s weaknesses section.
>
> ---
>
> **Assumption 2 is restrictive.** There is a misunderstanding of Assumption 2, and we apologise for the ambiguity in our original statement: the reviewer writes that our assumption is strong with respect to what most existing results in ICA do:
>
> > most existing results on nonlinear ICA assume only that the noises are jointly gaussian with environment-specific covariance matrices
> >
>
> **This is *exactly* how we intend Assumption 2**: the (understandable) confusion arises as we write that we rescale the sources of an initial environment to get the sources of an auxiliary environment. This is imprecise phrasing: given an initial environment with sources $\mathbf S^0 \sim \mathcal N(\mu^0, \Sigma^0)$, the auxiliary environment has sources $\mathbf S^e \sim \mathcal N(L_e \mu^0, L_e^2 \Sigma^0)$. So $\mathbf S^0$ and $\mathbf S^e$ are **independent**,  and the **equality between $\mathbf S_e$ and $L_e \mathbf S^0$ must be intended *in distribution***. To clarify this, in the updated version of the manuscript, we write:
>
> > **Assumption 2.** Each environment is obtained rescaling the covariance of $\mathbf S$, namely, $\mathbf S^i$ is *distributionally* equivalent to $L_i \mathbf S^0$ […]
> >
>
> ---
>
> **Experiments.** The reviewer requires experiments with larger graphs and a comparison with other ICA-based methods.
>
> - **Multivariate graphs experiments.** Experiments on multivariate graphs are added in **Appendix E.5** of the paper (please refer to the general response for details).
> - **Comparison with other nonlinear ICA-based methods**. We are only aware of two methods in the flavor of nonlinear ICA for causal discovery, which are from Reizinger et al. (2023) and Monti et al. (2019).
>     - **Monti et al. did not release their code.** It is thus impossible to perform experiments with their method.
>     - **Reizinger et al. is not suitable for multi-environment causal discovery**. They assume that the ICA model is *strongly identifiable* (i.e., no need for multiple environments), and train a VAE to get the causal graph from the encoder.
>
>     We are not aware of other freely accessible methodologies to compare to.
>
>
> ---
>
> **Questions**. We address the questions from the reviewer
>
> - *How is $L_i$ selected in numerical simulations?* We independently sample the diagonal coefficients from a uniform distribution between 2 and 5.
> - *With six and nine environments, are the values different across all environments?* Yes. This is because, otherwise, an environment doesn’t bring any additional information.
> - *Can we extend the theory without assuming fixed mechanisms across environments?* According to the Invariant Causal Predictions (ICP) paper, Peters et al. (2015), one can use general invariances between environments to infer the causal graph. In causal discovery, we would like edges of the graph to be invariant across environments: intuitively, this could be exploited according to the ICP principle. However, it’s unclear how to retain the independence between the sufficient number of environments and the dimension of the graph without the fixed mechanisms, a crucial part of our theory. This requires further research.
>
> ---
>
> **Summary.** We clarify the meaning of Assumption 2 in our paper and provide the required new experiments on multivariate graphs that further support our theory. As we address all the points raised in the review, **we kindly ask the reviewer to consider increasing their score. Thank you** for your time and consideration.

---

### Official Review · Reviewer_FXre · 2025-11-01

**Soundness:** 4
**Presentation:** 3
**Contribution:** 3
**Rating:** 8
**Confidence:** 4

**Summary:**

The authors establish that "the full causal graph of an invertible SCM with arbitrary nonlinear mechanisms is identifiable from the model’s distribution and data gathered from only two sufficiently variant environments", providing the first theoretical guarantee for complete causal graph recovery under a constant number of environments, in contrast to prior nonlinear identifiability results which require #environmental which scales linearly with data dimension. The analysis assumes Gaussian noise, yet the authors discuss how to relax this condition to a more general class of noise distribution (having at least one point with zero gradient of the likelihood, not necessarily at the mean as for Gaussian) . Their proof introduces a novel application of the recently highlighted duality between (non-linear) SCM and ICA identifiability, showing that causal discovery requires far less auxiliary information than nonlinear ICA. Empirical validation on synthetic data corroborates the theory, showing successful causal direction recovery in previously non-identifiable bivariate cases.

Overall assessment: The paper is well written and generally interesting. It builds on the recent duality between identifiability in non-linear SCM and ICA and suggests promising directions for advancing identifiability theory and scalable algorithms for high-dimensional, multi-environment causal discovery. While I enjoyed reading the paper, I would have appreciated more comments/remarks/analysis on limitations, for example for unobserved confounders and a more general class of distributions. Nevertheless, I do believe the paper brings useful insight to ICLR, and more specifically causality, research communities.

**Strengths:**

This work provides a first theoretical result toward identifiability of the full causal structure with as few as two environments with sufficient variability with respect to a base environment as specified in Assumption 5. The novel proof technique of exploiting the established duality of SCMs and ICA brings interesting methodology for subsequent analysis. The paper is well written and manifest a good balance between main body details and further explanations in the appendix. Empirical validation on synthetic data complements the theory, showing successful causal direction recovery in previously non-identifiable bivariate cases.

**Weaknesses:**

- Recent work shows that overcomplete ICA is beneficial in scenarios with unobserved confounders and a single environment. I think a discussion about the limitation of "extending this work with multiple environments" to causal diagrams with unobserved confounders could be of great interest to the community. For instance, what are the fundamental limitations to extend proof techniques to over complete non-linear ICA (more sources than variables)?

- The authors acknowledged the limitation of "Gaussian noise assumption" and commented on what other noise distributions to which their method extends (at least one point with zero gradient). I would have appreciated also a discussion about the fundamental limitation of the method to wider classes of distributions (i.e., from the perspective of necessary as opposed to sufficient conditions).

**Questions:**

- I think there is a typo in the equation of Proposition 1: The RHS should be $j\notin {\text{PA}}_{I}$, no?
- I think using an index for environment (say t or r)  different than that of variables (i) could make your presentation less confusing; see for example definition 4 and how it references equation (1). Also, a superscript for environment versus a subscript for variable could be defined in the notation section at the beginning of Section 3.1.
- I think that a closer mathematical explanation in the last part of Section 4.1 (Theorem 1 beyond Gaussianity) could be beneficial to inexperienced reader.

**Details Of Ethics Concerns:**

No unprofessional behavior is detected.
The paper builds on existing research in fundamental way, while to the best of my knowledge, cite key relevant research.

---

> ### Author Response · Authors · 2025-11-22
>
> We thank the reviewer for their time and constructive feedback. Also, we thank the reviewer for their appreciation of our work. Below, we address the points raised in the weaknesses section.
>
> ---
>
> **Discussion on overcomplete ICA.** The updated version of the manuscript contains a **detailed discussion** on possible extensions of our results to the overcomplete ICA setting (**Appendix F.2**). In summary, the fundamental issue with overcomplete ICA is that the causal mechanism in $\mathbf x = \mathbf f(\mathbf s)$ is not invertible, which prevents us from extending our results to nonlinear, confounded SCMs. In fact:
>
> 1. For $\mathbf f$ non-invertible, the change of variable formula over $p_X$ (eq. 4 and 5) does not hold.
> 2. As a consequence, Lemma 1 fails.
>
> Given that Lemma 1 is the fundamental result enabling Theorem 1, our proof technique can not generalize to overcomplete ICA in the nonlinear case. The only exception comes from linear, confounded SCMs of the form $x = As$: in that scenario, the covariance $\Sigma_X = A \Sigma_S A^T$ still carries information about the Gram matrix $AA^T$, which can be exploited to infer information about the causal graph.
>
> ---
>
> **Wider discussion on the noise assumption limitation.** The reviewer asks for
>
> > a wider discussion about the fundamental limitation of the method to a wider classes of distribution.
> >
>
> We agree that this is a valuable addition. In **Appendix F.1 of the updated manuscript,** we added
>
> 1. **An extended discussion on the mathematical details** that are mentioned in the paragraph “Theorem 1 beyond Gaussianity”; this also addresses Question 3 in the review.
> 2. **A detailed analysis of the fundamental challenges to extending our proof technique** to SCMs **with sources in the exponential family**. We choose the exponential family as this is (i) broad and (ii) very commonly assumed in the multienvironment ICA literature, hence in line with the rest of our work.
>
> ---
>
> **Questions.**
>
> - “I think there is a typo in the equation of Proposition 1”: Yes, thanks. We fixed that in the new version of the paper.
> - About the environment notation.
>     - We introduce the superscript notation in the *Notational remarks* paragraph.
>     - Regarding the change of letters for environments, we agree that this would be a good point; however,  we fear that implementing the change now would break the paper, given that it should be done by manual inspection over hundreds of occurrences: some indices are likely to be missed. As a result, we fear that this can reduce the clarity of the paper, rather than improving it.
> - A closer mathematical inspection of the paragraph “Theorem 1 beyond Gaussianity” is in the updated manuscript, **Appendix F.1**.
>
> ---
>
> We renew our gratitude to the reviewer for the positive and constructive feedback. In this rebuttal, we addressed all the concerns and questions they posed. **In light of these clarifications, if they believe the revised version of our paper is worth consideration for an oral or spotlight, we kindly ask them to consider raising their score.**

---

### Official Review · Reviewer_WHVU · 2025-11-08

**Soundness:** 3
**Presentation:** 3
**Contribution:** 3
**Rating:** 6
**Confidence:** 3

**Summary:**

This paper studies the identifiability of causal graphs when data come from multiple environments. The authors prove that for structural causal models (SCMs) with arbitrary nonlinear mechanisms and Gaussian noise, the full causal graph can be uniquely identified using data from only two distinct environments. The work builds on the theoretical duality between Independent Component Analysis (ICA) and causal discovery, offering a new identifiability result that links second-order statistics (Hessians of log-likelihoods) across environments to the Jacobian structure of the causal model. Synthetic experiments on bivariate models validate the theoretical claims.

**Strengths:**

I appreciate the authors’ clear and well-organized presentation of their work. The paper carefully states its assumptions, proofs, and detailed algorithms, demonstrating both theoretical rigor and practical insight. Most notably, it shows that full causal graph identifiability can be achieved without the number of required environments scaling with dimensionality—a highly appealing property for real-world applications, which I consider the paper’s most significant strength.

**Weaknesses:**

So far, the largest weakness of the paper is the narrowness of the experiment. They only test on simple two-variable synthetic examples. Although the authors acknowledge this as a limitation, I believe that experiments on multivariate settings are necessary. Without trying it on higher-dimensional problems or actual datasets, it's hard to gauge whether this really works in practice.

Another potential weakness lies in the assumption of Gaussian noise variables. At first glance, this may not seem overly restrictive for two reasons.

1. Noise is often regarded as the sum of many small unobserved factors, which should approximate normality according to the central limit theorem.

2. Even if the true noise is non-Gaussian, it can theoretically be expressed as an invertible transformation of a Gaussian variable, and such a mapping could be absorbed into the causal mechanisms without affecting the identifiability of the causal graph.

However, the non-Gaussian experiments reported in the appendix raise some concerns—the results are not entirely convincing. This suggests that, in practice, the Gaussianity assumption may indeed be a substantive limitation of the current work.

**Questions:**

1. For proposition~1, I believe there is a typo that should be $J_f^{-1}(x)_{ij} \not=0$.

2. Can the authors discuss the potential reasons why including more domains causes performance to decrease? I agree that in theory, two domains are enough. However, for nearly all CRL papers, we benefited from additional domains. Is that an optimization issue?

---

> ### Author Response · Authors · 2025-11-22
>
> We thank the reviewer for their constructive feedback and the time dedicated to our paper. Below, we address the concerns raised in their comments.
>
> ---
>
> **Experiments**. The main point in the review is that experiments are limited to bivariate graphs. Experiments on multivariate graphs are added and analysed in Appendix E.5 of the paper (please refer to the general response of this rebuttal for details).
>
> ---
>
> **Experimental results on non-Gaussian noise**. The reviewer writes that “the non-Gaussian experiments reported in the appendix raise some concerns”. We understand this point of view, but we do not share the concern: the goal of the experiments is simply that of validating our theory; with this in mind, we design Algorithm 1 as a naive algorithmic implementation of the steps in the proof of Theorem 1. As a consequence, the method inherits the dependency on the Gaussianity assumption and is expected to fail when this is violated. In summary, **our empirical results simply reflect the dependency of Theorem 1 on Gaussianity,** rather than pointing to fundamental algorithmic limitations for other noise distributions.
>
> As we discuss in the paper, an algorithmic contribution that fully leverages the results of our theory is left for the future.
>
> ---
>
> **Questions**
>
> 1. Typo in proposition 1: good point; we fixed that in the current version of the paper. Thanks!
> 2. Looking at **Figure 1** in the main paper, we find that for different mechanism types adding environments sometimes improves and sometimes decreases the mean SHD. Moreover, differences are always within error bars, meaning that they are not statistically significant. Similar comments hold for Figures 5 and 6.
> Interestingly, **Figure 4** shows that when identifiability is harder (i.e., when the score vanishes at some point, but Gaussianity is violated), additional environments are beneficial for inference. This aligns with what one would expect extrapolating from the CRL literature, given that more environments mean more constraints on the solution.
>
> ---
>
> We renew our appreciation for the reviewer’s work and feedback on our paper. **Given that we address the main concern by adding experiments on larger graphs**, and in light of the clarifications in this rebuttal, **we kindly ask the reviewer to consider increasing their rating**. In case of doubts, we remain available for further clarifications.

---

### Author Response · Authors · 2025-11-22
**General response**

We thank the reviewers for the thorough and constructive feedback. We are happy that the contributions of our work are enthusiastically appreciated:

- **All reviewers** praise the **clarity** of exposition (scores ranging between 3 and 4).
- R eLzQ, R FXre, R WHVU, R ZrxN  **praise the novelty** of “***the first theoretical guarantee** for complete causal graph recovery under a constant number of environments”* (R FXre)
- The work provides **extremely important insights** (R eLzQ) and **shows promising directions to advance identifiability theory** (R FXre).
- Our findings are **highly appealing for real-world applications** (R WHVU)

The **one main concern shared** by R ZrxN and R WHVU (which is also a minor concern to R 5jiF) is the absence of experiments on multivariate graphs. These are now added to Appendix E.5 of the paper, and summarised below. All the remaining points of the reviews are addressed in the individual responses.

## Multivariate graphs experiments

In Appendix E.5 of the paper, we add experiments with multiple nodes, as required by R ZrxN, 5jiF, WHVU. We summarise our additions below.

**Linear multivariate experiments.** In the linear setting, we replace the Hessian *Stein gradient estimator* with a simple covariance estimator from numpy. This scales much better to higher dimensions, while being equivalent to the (inverse of) the Hessian estimator in the population limit. We find that on graphs with 10, 20, and 50 nodes, **our method, compared to a random baseline, reduces the error by 75% on 10 nodes and 30% on 50 nodes**.

**Nonlinear multivariate experiments.** In the nonlinear setting, we report experiments on 5 nodes, as required by R 5jiF and ZrxN. Notably, previous ICA-based causal discovery methods (Reizinger et al., 2023) generally even fail to converge to a solution for graphs with $5$ nodes. In contrast, we find that, compared to a random baseline, **our method reduces the error by a value between 25% and 30%** (across different nonlinear mechanisms).

In both linear and nonlinear settings, the reported accuracy refers to inference in 3 environments. Adding environments (6 or 9) does not significantly change the accuracy, in line with our theory.

---

## List of changes in the updated paper

All changes in the paper are highlighted in blue.

- Appendix E.5: multivariate graphs experiments (required by R ZrxN, R 5jiF, and R WHVU)
- Appendix G.5: a list of real-world research problems in causality where modelling assumptions align with those in our theory (prompted by comments from R 5jiF)
- Appendix F.1: a detailed mathematical discussion of the content of “Theorem 1 beyond Gaussianity” paragraph (required by R FXre)
- Appendix F.1: a detailed discussion on wider classes of noise distributions (required by R FXre)
- Appendix F.2: relation with overcomplete ICA (required by R FXre)
- Clarification of Assumption 2 (prompted by comments from R ZrxN)
- Abstract rewording (required by eLzQ)
- Missing citations (required by eLzQ)
- Fix of typo in Proposition 1 (multiple reviewers)
- Other minor changes prompted by the reviewers

---

### Author Response · Authors · 2025-11-27

Dear all,

We noted that the majority of reviewers did not engage in a discussion with us. As the rebuttal end period approaches, we are eager to clarify any remaining doubts about our work. Once again, we thank you for your time and consideration of our work.

Kind regards,
the Authors

---

### Author Response · Authors · 2025-12-01
**Summary for the Area Chair**

Dear Area Chairs and Reviewers,

We express our sincere gratitude for the effort in evaluating our paper. Given the unusual circumstances, we would like to outline how we addressed reviewers’ concerns in the rebuttal, to facilitate the Area Chair in evaluating our work. Also, we note that, since the initial reviews, the paper was strongly endorsed by R FXre and R eLzQ (rating the paper 8 and 10).

---

In what follows, we summarise the main concerns from R WHVU, R ZrxN, R 5jiF, R FXre, and how they are addressed in the rebuttal and discussion.

- R WHVU, R ZrxN, and R 5jiF require more experiments on multivariate graphs. In Appendix E.5 we added the required experiments with up to 50 nodes for linear models, and 5 nodes for nonlinear models; the target of 5 nodes is in line with the requests of R 5jiF and R ZrxN.
- R 5jiF requires clarifications on Assumption 2, which we provide in the rebuttal.
- R 5jiF argues that our work goes against the goals of interventional causal discovery.
    - After the rebuttal, the reviewer agrees that our work aligns with what they presented as one of the goals of interventional causal discovery in their review. However, they express reservations that (i) we do not present a theory of identifiability of equivalence classes under weaker assumptions, and (ii) that our proof technique is not novel.
    - In the response, we motivate why we disagree with their second claim.
- R ZrxN argues that our assumption 2 is too strong: as we discuss in the rebuttal, this concern (understandably) arises due to a lack of clarity in our original text. We rephrased the text of our Assumption 2, which is now aligned with the weaker formulation the reviewer pointed to—that is how our assumption was originally intended.
- R FXre asks for a discussion connecting our work with overcomplete ICA, and details about possible extensions of our theory beyond Gaussianity. We added the required discussion in Appendices F.1 and F.2.

We renew our gratitude to the Area Chair and Reviewers for committing to our work.

---

### Meta-Review · Area_Chair_MUxf · 2026-01-07

**Summary:**

This paper introduces Regularized Latent Dynamics Prediction, a representation learning method for zero-shot reinforcement learning that uses self-supervised latent next-state prediction augmented with orthogonality regularization. The method is evaluated across multiple benchmarks (16 DMC tasks, 45 Humanoid tasks, 6 D4RL tasks) with up to 358-dimensional observations and 69-dimensional actions, demonstrating competitive performance compared to state-of-the-art Behavioral Foundation Model methods (FB, PSM, Laplace, HILP), and in low-coverage settings.
All four reviewers acknowledged the paper's strengths: theoretical soundness connecting latent dynamics to successor measures, simplicity and ease of implementation compared to existing methods, comprehensive experimental evaluation, and strong performance in challenging low-coverage scenarios.

**Reviewer Concerns:**

Addressed Concerns:
1.	Clarifying differences from existing methods (2m5w): Authors added Appendix A.2 explaining existing SOTA methods and their relationship to RLDP.
2.	Writing and clarity issues (Eidy):
○	Method section updated to clearly specify which parameters are optimized for each loss
○	Writing improvements for paragraph formatting and typos throughout
○	Overclaimed statement about "minimizing prediction error for successor measures" corrected
3.	Missing experimental details (2m5w, Eidy):
○	Added variance statistics to Figure 3 with Mann-Whitney U tests
○	Added Figure 9 with IQM aggregation for better readability
○	Added Figure 6 showing reference baseline for cosine similarity to justify "mild collapse" claim
○	Added D4RL medium-replay datasets (Table 10)
○	Added random feature baseline (Table 7)
○	Added gridworld experiments with qualitative visualizations (Appendix A.8)
4.	Theory-practice alignment (GfJN): Authors clarified that RLDP learns representations independently from successor measure estimation, and that Lemma A.1 provides an upper bound rather than claiming perfect successor measure learning.
5.	Applicability questions (GfJN): Authors added Universal Successor Feature experiment (Table 8) demonstrating broader applicability of learned representations.

Remaining Concerns:
1.	Limited improvement on ExoRL benchmark (GfJN): While authors note the benchmark is "saturated to some extent," the competitive rather than dominant performance on this standard benchmark remains. However, authors appropriately expanded evaluation to 45 additional Humanoid tasks and D4RL to demonstrate RLDP's merits more comprehensively.
2.	Real-world validation (Th7h): All experiments remain in simulated environments. Authors provide reasonable discussion (Appendix A.4.4) about expected challenges but no empirical validation on real-world data.

**Reviewer Scores:**

Original scores:
●	Reviewer 2m5w: 8 (accept, good paper - poster)
●	Reviewer Eidy: 6 (marginally above acceptance threshold)
●	Reviewer Th7h: 6 (marginally above acceptance threshold)
●	Reviewer GfJN: 6 (marginally above acceptance threshold)

Expected post-rebuttal scores: Based on the comprehensive rebuttal addressing all major concerns with substantial new experiments and clarifications, I anticipate:
●	Reviewer 2m5w: Likely to maintain 8 (all concerns addressed)
●	Reviewer Eidy: Likely to increase to 8 (major clarity and experimental concerns thoroughly addressed)
●	Reviewer Th7h: Likely to maintain 6 (concerns addressed, though real-world validation remains future work)
●	Reviewer GfJN: Likely to maintain 6 (USF experiment added, theory-practice mismatch clarified)

---

> ### Public Comment · ~Francesco_Montagna2 · 2026-03-18
> **Wrong meta-review**
>
> Dear Area Chair,
>
> We note that there must have been an error in the publishing of the meta-review, which clearly refers to another work.

---

### Decision · Program_Chairs · 2026-01-26

Accept (Poster)